# Comparison of dimension reduction techniques in the analysis of mass spectrometry data

Sini Isokääntä[1], Eetu Kari[1,a], Angela Buchholz[1], Liqing Hao[1], Siegfried Schobesberger[1], Annele Virtanen[1], Santtu Mikkonen[1,2]

[1] Department of Applied Physics, University of Eastern Finland, Kuopio, 70210, Finland
[2] Department of Environmental and Biological Sciences, University of Eastern Finland, Kuopio, 70210, Finland
[a] currently at: Neste Oyj, Espoo, Finland

*Correspondence to*: S. Isokääntä (sini.isokaanta@uef.fi)

**Abstract.** Online analysis with mass spectrometers produces complex data sets, consisting of mass spectra with a large number of chemical compounds (ions). Statistical dimension reduction techniques (SDRTs) are able to condense complex data sets into a more compact form while preserving the information included in the original observations. The general principle of these techniques is to investigate the underlying dependencies of the measured variables, by combining variables with similar characteristics to distinct groups, called factors or components. Currently, positive matrix factorization (PMF) is the most commonly exploited SDRT across a range of atmospheric studies, in particular for source apportionment. In this study, we used 5 different SDRTs in analysing mass spectral data from complex gas- and particle phase measurements during laboratory experiment investigating the interactions of gasoline car exhaust and α-pinene. Specifically, we used four factor analysis techniques: principal component analysis (PCA), positive matrix factorization (PMF), exploratory factor analysis (EFA), and non-negative matrix factorization (NMF), as well as one clustering technique, partitioning around medoids (PAM).

All SDRTs were able to resolve 4-5 factors from the gas phase measurements, including an α-pinene precursor factor, 2-3 oxidation product factors and a background/car exhaust precursor factor. NMF and PMF provided an additional oxidation product factor, which was not found by other SDRTs. The results from EFA and PCA were similar after applying oblique rotations. For the particle phase measurements, four factors were discovered with NMF: one primary factor, a mixed LVOOA factor, and two α-pinene SOA derived factors. PMF was able to separate two factors: SVOOA and LVOOA. PAM was not able to resolve interpretable clusters due to general limitations of clustering methods, as the high degree of fragmentation taking place in the AMS causes different compounds formed at different stages in the experiment to be detected at the same variable. However, when preliminary analysis is needed, or isomers and mixed sources are not expected, cluster analysis may be a useful tool as the results are simpler and thus easier to interpret. In the factor analysis techniques, any single ion generally contributes to multiple factors, although EFA and PCA try to minimize this spread.

Our analysis shows that different SDRTs put emphasis on different parts of the data, and with only one technique some interesting data properties may still stay undiscovered. Thus, validation of the acquired results either by comparing between

different SDRTs or applying one technique multiple times (e.g. by resampling the data or giving different starting values for iterative algorithms) is important as it may protect the user from dismissing unexpected results as "unphysical".

## 1    Introduction

Online measurements with mass spectrometers produce highly complex data comprised of hundreds of detected ions. High resolution mass spectrometer enables identification of the elemental composition of these ions revealing chemical composition information about the sample. However, even with the highest resolution, mass spectrometers are not able to resolve isomers. Instead the large number of identified ions can make data interpretation challenging due to the sheer number of variables. Different statistical dimension reduction techniques (SDRT) were developed to compress the information from complex composition data into a small number of factors, which can be further interpreted by their physical or chemical properties. In other words, these methods are used to understand the underlying relationships of the measured variables (i.e., detected ions). Principal component analysis (PCA), which was introduced already in the beginning of the 20[th] century by Karl Pearson is probably the first SDRT, even if the modern formulation of PCA was introduced decades later (Pearson., 1901; Hotelling, 1933). In atmospheric studies, the most exploited method, especially in the analysis of long time series of aerosol mass spectrometer (AMS) data, is positive matrix factorization (PMF) developed in the mid-1990s (Paatero and Tapper, 1994). Other SDRTs that are widely applied in different fields of science for the analysis of multivariate data include PCA and exploratory factor analysis (EFA), which are popular especially in medical and psychological studies (Raskin and Terry, 1988; Fabrigar et al., 1999). In atmospheric studies, the latter methods have not gained as wide popularity, but a few examples still exist. Customized PCA was applied to organic aerosol data collected from Pittsburgh in 2002 (Zhang et al., 2005) and a more traditional version of PCA was used to analyze Chemical Ionization Reaction Time-of-Flight Mass Spectrometer (CIR-ToF-MS) and compact Time-of-Flight Aerosol Mass Spectrometer (cToF-AMS) data acquired in smog chamber studies during several measurement campaigns (Wyche et al., 2015). Additionally, EFA and PCA have been applied in several source apportionment studied in the environmental science fields (Pekey et al., 2005; Sofowote et al., 2008), and a recent study on plant VOC emissions applied EFA to separate effects of herbivory induced stress from the natural diurnal cycle of the plants (Kari et al., 2019a). Very much like PMF, non-negative matrix factorization (NMF) is one of the most used methods in the analysis of DNA microarrays and metagenes in computational biology (Brunet et al., 2004; Devarajan, 2008), but NMF has also been applied in atmospheric studies (Chen et al., 2013; Malley et al., 2014).

Comparisons between the performance of some of the SDRTs presented in this paper already exists, but due to the popularity of PMF, other methods are not applied as widely in atmospheric studies. As EFA and PCA are rather similar methods, and they have also existed many decades, multiple comparisons between them exists, especially in the medical and psychological research fields (see. e.g. Kim, 2008). The introduction of PMF has also inspired comparison studies between PMF and EFA (Huang et al., 1999), and PMF and PCA were already shortly compared upon publication of PMF as the positivity constraints were presented as an advantage over PCA (Paatero and Tapper, 1994). Although PMF has been shown to be a very powerful

tool in the analysis of environmental AMS data from field studies (e.g. Ulbrich et al., 2009; Zhang et al., 2011; Hao et al., 2014, Chakraborty et al., 2015), it has not been applied as widely in laboratory and smog chamber research (Corbin et al., 2015; Kortelainen et al., 2015; Tiitta et al., 2016; Koss et al., 2020). Latest studies have applied PMF also to chemical ionization mass spectrometry data (Yan et al., 2016; Massoli et al., 2018; Koss et al., 2020) which is able to resolve more oxidized compounds. The special conditions in lab experiments (sharp change at the beginning of experiments, e.g., switching on UV-lights) present an additional test scenario, as PMF has been mostly used for field measurement data sets where the main focus is often in the long-term trends and real changes in factors are expected to be more subtle than e.g. the variations in the noise in the data. In addition, field measurements commonly yield very large data sets, including thousands of time points, whereas laboratory experiments may be much shorter. Recently, scientists from atmospheric studies have been motivated to test and adapt other techniques and algorithms to reduce the dimensionalities of their data, in addition to the more "traditional" version PMF introduced in the 90s. For example, Rosati et al. (2019) introduced correlation based technique for multivariate curve analysis (similar to NMF) in their analysis of α-pinene ozonolysis. Cluster analysis has been applied in a few studies. Wyche et al., 2015) applied hierarchical cluster analysis (HCA) to investigate the relationships between terpene and mesocosm systems. In the study from Äijälä et al. (2017), they combined PMF and k-means clustering to classify and extract the characteristics of organic components. In addition, very recent paper by Koss et al. (2020) also compared the dimension reduction abilities of HCA and gamma kinetics parametrization to PMF when studying mass spectrometric data sets.

In our study we chose a set of SDRTs having fundamental differences. For example, PMF usually splits one ion into several factors, whereas most of clustering techniques assign one ion into one cluster only. If isomers with the same chemical composition but different functionality are expected, splitting of ions into several factors might be preferred. On the other hand, clustering might be more suitable for a more simplified or preliminary approach (as it is computationally less demanding), or when the chemical compounds in the data are already known or if strict division between variables is preferred. In this study, we validate the usability of the chosen SDRTs in laboratory studies for two different mass spectrometer devices, PTR-ToF MS and AMS (gas- and particle phase composition), and different data sizes due to different measuring periods and time resolutions. Further, we examine the performance of the SDTRs when the data includes large and rapid changes in the composition.

## 2    Experimental data

The datasets investigated in this study were gathered during experiments conducted as part of the TRACA campaign at the University of Eastern Finland. A detailed description of the experimental setup and reaction conditions can be found in Kari et al., 2019b). Briefly, the measurement setup consisted of a modern gasoline car (VW Golf, 1.2 TSI, Euro 6 classification) which was driven at a constant load of 80 kilometres per hour after a warm up period with its front tiers in a dynamometer. The exhaust was diluted using a two-stage dilution system and fed into a 29 m$^3$ collapsible environmental PTFE chamber ILMARI (Leskinen et al., 2015). For the experiment investigated in this study, α-pinene (~ 1 µL, corresponding to 5 ppbV)

was injected into the chamber to resemble biogenic VOCs in typical suburban area in Finland. Atmospherically relevant conditions were simulated by adding $O_3$ to convert extra NO from vehicle emissions to $NO_2$ and adding more $NO_2$ to the chamber if needed. With these additions, atmospherically relevant VOC-to-NOx (~ 7.4 ppbC/ppb) and $NO_2$-to-NO ratios were achieved to resemble the typical observed level in suburban areas (National Research Council, 1991). Chamber temperature

was held constant at ~20°C and relative humidity was adjusted to ~50% before the start of the experiment. Blacklight (UV-A) lamps with a light spectrum centred at 340 nm were used to form OH radicals from the photolysis of $H_2O_2$. The start of photooxidation by turning on the lamps is defined as experiment time 0 in the following. Vertical dashed lines in the figures indicate α-pinene injection and the start of photo-oxidation, respectively. A short summary of the experimental conditions and the behaviour of the injected α-pinene as a time series are shown in the SI material (Sect. S1)

Volatile organic compounds (VOCs) in the gas phase were monitored with a proton-transfer-reaction time-of-flight mass spectrometer (PTR-TOF MS 8000, Ionicon Analytik, Austria, hereafter referred to as PTR-MS). Typical concentration for few example VOCs in the mid-way of the experiment were 2 µm/m$^3$ for toluene, 0.2 µm/m$^3$ for TMB (trimethylbenzene) and 1.7 µm/m$^3$ for $C_4H_4O_3$. Detailed setup, calibration procedure and data analysis of the used high resolution PTR-MS have been explicitly presented in Kari et al. (2019b). In the campaign, the high mass resolution of the instrument (>5000) enabled the

determination of the elemental compositions of measured VOCs. The instrumental setting intended to minimize the fragmentation of some compounds, so the quantitation of the VOCs was possible. The chemical composition of the particle phase of the formed SOA was monitored with a soot particle aerosol mass spectrometer (SP-AMS, Aerodyne Research Inc., USA, hereafter referred to only as AMS, Onasch et al., 2012). In brief, the SP-AMS was operated at 5 min saving cycles, alternatively switching between the electron ionization (EI) mode and SP mode. In EI mode, the V-mode mass spectra were

processed to determine the aerosol mass concentration and size distribution. The mass resolution in the mode reaches ~2000. The SP-mode mass spectra were used to obtain black carbon concentration. As the used chamber was a collapsible bag, the volume of the chamber decreased over time due to the air taken by the instruments. For the experiment investigated in this study, both gas- and particle phase data were analyzed with all SDRTs (Sect. 4.1 and 4.2). However, due to the small data size for the particle phase, not all SDRTs were applicable.

In contrast to the PTR-MS data used in Kari et al. (2019b), we did not apply baseline correction to the data. Overestimation of the baseline correction may cause some of the ions with low signal intensity to have negative "concentration", which is not physically interpretable. Also, negative data values cause problems for some SDRTs, as e.g. PMF and NMF need a positive input data matrix. In addition, SDRTs should be able to separate background ions into their own factor, meaning it is not mandatory to remove them before applying SDRTs. This approach will cause some bias to the absolute concentrations of the

ions and resulting factors, but as we are more interested in the general division of the ions to different factors, and their behaviour as a time series when comparing the SDRTs, it does not significantly affect our interpretation of the results. All recommended corrections (including baseline subtraction, Ulbrich et al., 2009) were applied to the AMS data. As the processed AMS data is always the difference between the measured signal with and without particles, negative values are possible if the particle free background was elevated. In the investigated dataset, only few data points exhibited slightly negative values.

Thus, it was possible to set these data points to a very small positive value ($1\cdot10^{-9}$) to enable the analysis with SDRT methods without a significant positive bias in the data. In addition, as the main focus of our study was to compare the performance of the different SDRTs with different types of mass spectra, instead of detailed analysis of the chamber experiment, we have also included the pre-mixing period during the α-pinene (i.e. t < 0) injection into our analysis.

## 3    Dimension reduction techniques

### 3.1    Factorization techniques

#### 3.1.1    Principal Component Analysis (PCA)

PCA is a statistical procedure where the variables are transformed into a new coordinate system. The first principal component accounts for the most variance of the observed data and each succeeding component has then the largest possible variance with the limitation that the component must preserve orthogonality to the preceding component. In other words, PCA seeks correlated variables and attempts to combine them into a set of uncorrelated variables, i.e., principal components, which include as much as possible of the information that was present in the original observations (Wold et al., 1987; Morrison, 2005; Rencher and Christensen, 2012; Tabachnick and Fidell, 2014). The principal components are often described by a group of linear equations, where e.g. the first principal component $c_1$ (table of the used mathematical symbols and notations is presented in the appendix) can be presented as

$$c_1 = a_{11}y_1 + a_{12}y_2 + \cdots + a_{1m}y_m, \tag{1}$$

where the coefficients $a_{1j}$ ($j = 1, \ldots, m$) are normalized characteristic vector elements assigned to the specific characteristic root of the correlation matrix $\mathbf{S}$, and $y_i$ ($i = 1, \ldots, m$) are the centred variables (Morrison, 2005; Rencher and Christensen, 2012). As the responses in the first principal component have the largest sample variance, $s_{y_1}^2$, for all normalized coefficient vectors applies

$$s_{y_1}^2 = \sum_{i=1}^{m} \sum_{j=1}^{m} a_{1j} a_{1j} s_{ij} = \boldsymbol{a}_1^T \mathbf{S} \boldsymbol{a}_1, \tag{2}$$

where $\boldsymbol{a}_1^T \boldsymbol{a}_1 = 1$ (Morrison, 2005). The number of principal components is equal to the number of variables (m) in the data minus one, and $p$ components are selected to interpret the data. It should be noted, however, that Eq. (1) describes the theory behind the PCA model, not the actual calculation process, which is described below. Thus, for example, centring of variables is not required. To find the principal components, either eigenvalue decomposition (EVD) or singular value decomposition (SVD) can be used. Mathematical formulation of EVD and SVD can be found from Golub and Van Loan, 1996). EVD is applied to the correlation or covariance matrix $\mathbf{S}$ whereas SVD can be applied also to the observed data matrix directly. Often, due to this difference, SVD is considered as its own method instead of describing it as a variation of PCA. Here, however, it is referred to as SVD-PCA. In our study we applied EVD-PCA to the correlation matrix (calculated from unscaled data matrix) and SVD-PCA was applied to the data matrix without and with the scaling (centred and scaled by their standard deviations). In addition, the acquired eigenvectors and vectors corresponding to the singular values were scaled by the square root of the

eigenvalues/singular values to produce loading values (i.e., contribution of a variable to a component) more similar to those obtained in exploratory factor analysis (EFA). The PCA analysis was performed in R statistical software with the addition of the psych-package (Revelle, 2018; R Core Team, 2019).

The acquired principal components can be rotated to enhance the interpretability of the components. Rotations can be performed in orthogonal or oblique manner, where the orthogonal methods preserve the orthogonality of the components, but oblique methods allow some correlation. However, rotation of the principal components does not produce another set of principal components but merely components. By original definition (Hotelling, 1933), only presenting unrotated solution is considered as principal component analysis, but later formulations allow also orthogonal rotations (Wold et al., 1987). Though there are no computational restrictions for applying oblique rotation on components, the restriction is only definitional, as the original principal components were presented as orthogonal transformation. In any case, rotated solutions do not fulfil the assumption of principal ordering of components. In this study, orthogonal Varimax rotation, which maximises the squared correlations between the variables, and oblique Oblimin rotation were used to increase large loading values and suppress the small ones to simplify the interpretation (Kaiser, 1958; Harman, 1976).

Multiple ways exist to calculate the PCA component scores (i.e., component time series). In general, the components scores are calculated as

$$\mathbf{F} = \mathbf{XB} \tag{3}$$

where $\mathbf{X}$ includes the analyzed variables (often centred and scaled by their standard deviations), and $\mathbf{B}$ is the component coefficient weight matrix (Comrey, 1973). One simple way to calculate the component scores is to use the component loading values directly as weights. This approach is often referred to as a sum score method. Depending on the application, the loadings can be used as they are, they can be dichotomized (1 for loaded and 0 for not loaded), or they can be used as they are but suppressing the low values by some threshold limit. We applied the latter method here, as the dichotomized loadings (i.e., one ion stems only from one source or source process) seldom describe true physical conditions in nature. In a case when the data items are on the same unit, the data may be used without standardization (Comrey, 1973). As this is the case in all of our respective data sets (concentration units for PTR-MS data are ppb and µg/m³ for AMS), the scores are calculated without standardising the data matrix to achieve more interpretable component time series. This applies for both EVD-PCA and SVD-PCA component scores. Very small loading values (absolute value less than 0.3) are suppressed to zero to enhance the separation of ions between the components. The limit of 0.3 was selected, as this is often given as a reference value for insignificant loadings (see. e.g. Field, 2013; Izquierdo et al., 2014). The components from PCA in the results sections are labelled with CO.

### 3.1.2 Exploratory Factor Analysis (EFA)

Similar to the EVD-PCA which takes an advantage of the correlations between the original variables, EFA generates the factors trying to explain the correlation between the measured variables (Rencher and Christensen, 2012). For a data matrix $\mathbf{X}$ having $m$ variables (ions) and $n$ observations (time points), the EFA model expresses each variable $\mathbf{y_i}$ ($i = 1, 2, \ldots , m$) as a linear

combination of latent factors $f_j$ ($j = 1, 2, \ldots, p$), where $p$ is the selected number of factors. So, for example for variable $y_1$ the EFA model can be presented as

$$y_1 - \bar{y}_1 = \lambda_{11}f_1 + \lambda_{12}f_2 + \cdots + \lambda_{1p}f_p + \varepsilon_1, \tag{4}$$

where the residual term $\varepsilon_1$ accounts for the unique variance, $m$ is the number of factors, $\bar{y}_1$ is the mean of the variable $y_1$, and

$\lambda_{ij}$ are the elements from the loading matrix $\lambda$ and serve as weights to show how each factor $f_j$ is contributing for each variable $y_i$ (Morrison, 2005; Rencher and Christensen, 2012). As for the PCA explained in the previous section, the Eq. (4) here describes the form of an EFA model based on literature, not the direct calculation in the algorithm. Thus, no scaling (what e.g. the subtraction of the mean $\bar{y}_1$ from $y_1$ in Eq. (4) essentially is) is applied here.

Different methods exist to calculate the factorization in EFA. In this study, principal axis factoring (hereafter pa-EFA) and

maximum likelihood factor analysis (hereafter ml-EFA) were selected due to their suitability for our data (explained in more detail in section 3.3). In pa-EFA, the minimizable function $F_1$ can be presented as

$$F_1 = \sum_i \sum_j \left(S_{ij} - R_{ij}\right)^2, \tag{5}$$

where $S_{ij}$ is an element of the observed correlation matrix $\mathbf{S}$ and $R_{ij}$ is the element of the implied correlation matrix $\mathbf{R}$. Maximum likelihood factor analysis, on the other hand, minimizes the function $F_2$

$$F_2 = \sum_i \sum_j \frac{\left(S_{ij} - R_{ij}\right)^2}{u_i^2 u_j^2}, \tag{6}$$

where the variances $u_i$ and $u_j$ for the variables $i$ and $j$ are considered. In other words, ml-EFA assigns less weight to the weaker correlations between the variables (de Winter and Dodou, 2012; Rencher and Christensen, 2012; Tabachnick and Fidell, 2014). In contrast to PCA, rotations are a recommended practice before interpreting the results in EFA, and the unrotated factor matrices are rarely useful (Osborne, 2014). Oblique Oblimin rotation was used to rotate the EFA factors. Orthogonal Varimax

rotation was also tested, but as the orthogonality assumption for the factors is rather stringent for this type of chemical data, and it produced uninterpretable factors, those results are omitted. EFA was run in R Statistical Software with an addition of the psych-package (Revelle, 2018; R Core Team, 2019) and the factor scores were calculated as described above for PCA. The factors from EFA in the results are labelled as FE.

### 3.1.3    Positive Matrix Factorization (PMF)

PMF is a bilinear model and can be presented as

$$\mathbf{X} = \mathbf{GF} + \mathbf{E} \tag{7}$$

where the original data matrix $\mathbf{X}$ is approximated with matrices $\mathbf{G}$ and $\mathbf{F}$, and $\mathbf{E}$ is the residual matrix, i.e., the difference between observations in $\mathbf{X}$ and the approximation $\mathbf{GF}$. After the factorization rank is defined by the user, Eq. (7) is solved iteratively in least squares sense. The values of $\mathbf{G}$ and $\mathbf{F}$ are constrained to be positive, and the object function $Q$ is minimized

(Paatero, 1997)

$$Q = \sum_{i=1}^{m} \sum_{j=1}^{n} \left(\frac{E_{ij}}{\mu_{ij}}\right)^2, \tag{8}$$

The term $\mu_{ij}$ in Eq. (8) includes the measurement uncertainties for the observation matrix **X** at time point $i$ for ion $j$. Originally, **μ** was calculated as the standard deviations of **X**, but other error types have also been used (Paatero and Tapper, 1994; Paatero, 1997; Yan et al., 2016). As apparent from Eq. (8), the measurement errors ($\mu_{ij}$) act as weighting values for the data matrix.

Thus, the chosen error scheme can have significant impact on the behaviour of $Q$.

To test this, different error schemes were investigated. The standard deviation values alone were not used as an error as the data includes fast concentration changes due to the sudden ignition of photo-oxidation which causes the standard deviations to be systematically too large. But as a reference, the standard error of the mean (the standard deviations of the ion traces divided by the square root of the number of observations, i.e., length of the ion time series) was used as an error for both PTR-MS and

AMS data. It considers that measurements with less observations contain more uncertainty. These error values are constant for each ion throughout the time series and do not change with signal intensity. This type of error is labelled here as static error. In addition, a minimum error estimate was applied, as suggested by Ulbrich et al., 2009). Determination of the minimum error for PTR-MS is presented in the SI section S2.1 and for AMS in section S2.2.

Additionally, an error following the changes in ion concentration was constructed for PTR-MS data by applying a local

polynomial regression to smooth the ion time series (R-function loess, Cleveland et al., 1992). From the regression fit the residuals were calculated and the running standard deviation from the residuals was used as an error. Again, the minimum error was applied here. This error is referred hereafter as signal following error. For AMS, we also applied a standard error that is frequently used by the AMS community. The standard AMS error consist of minimum error related duty cycle of the instrument and counting statistics following the Poisson distribution (Allan et al., 2003; Ulbrich et al., 2009). . Shortly, the

standard AMS error for signal $I$ can be formulated as

$$\mathbf{I}_{err} = \alpha \sqrt{\frac{I_O + I_C}{t_s}}, \tag{9}$$

where $\alpha$ is an empirically determined constant (here $\alpha = 1.2$, generated by the AMS analysis software PIKA, http://cires1.colorado.edu/jimenez-group/ToFAMSResources/ToFSoftware/index.html, last visit on 9.9.2019), $I_O$ and $I_C$ are the raw signal of the ion of particle beam (ions s$^{-1}$) for the chopper at open and closed position respectively, and $t_s$ is the

sampling time at a particular m/z channel (s).

Examples of the used error values for PTR-MS and AMS data are presented in SI at the end of sections S2.1 and S2.2, and the signal-to-noise ratios for different error matrices are reported in Sect. S2.3. In contrast to the suggested best practice (Paatero and Hopke, 2003), we did not downweighed any ions in our data sets. This approach was used in order to give each SDRT equal starting point for the analysis, as e.g. for NMF or PCA similar downweighing is not possible, as we do not have any

error estimates to calculate the signal-to-noise ratios in similar manner. However, to avoid misguiding the reader to omit recommended data pre-processing practice for PMF, we also tested PMF with downweighing. This, as expected, did not change

our results significantly, but we acknowledge it should be indeed applied if aiming for a more detailed chemical interpretation of the PMF factors.

Often, constraining the values to be positive is not enough to produce a unique PMF solution for Eq. (7). This can be assessed by applying rotations, as in EFA and PCA. The rotations in PMF are controlled through the Fpeak-parameter in which the changes produce new **G** and **F** matrices by holding the $Q$ value approximately constant (Paatero et al., 2002). In this study, rotations with Fpeak = (-1, -0.5, 0, 0.5, 1) were tested. PMF analyses were conducted in Igor Pro 7 (WaveMetrics, Inc., Portland, Oregon) with the PMF Evaluation Tool (Ulbrich et al., 2009). The acquired results were further processed in R Statistical Software (R Core Team, 2019). The factors from PMF are labelled as FP in the results,

### 3.1.4    Non-negative Matrix Factorization (NMF)

Non-negative matrix factorization was introduced to the wider public after Lee and Seung presented their application of NMF to facial image database in Nature (Lee and Seung, 1999). The method has since gained popularity, and it has been used in various scientific fields including, e.g., gene array analysis (Kim and Tidor, 2003; Brunet et al., 2004). As in PMF, the NMF solution is constricted to positive values only to simplify the interpretation of the results, and in principle, both of these methods attempt to solve the same bilinear equation. In contrast to PMF, the algorithms in NMF do not require an error matrix as an input and it makes therefore no assumptions of the measurement error, and therefore we present NMF here as a separate method from PMF.

In general, the mathematical formulation of NMF is similar to the one presented for PMF in Eq. (7) and can be presented as

$$\mathbf{X} \sim \mathbf{WH}, \tag{10}$$

where **X** is the positive data matrix ($n$ x $m$) and **W** and **H** are the nonnegative matrices from the factorization with sizes $n$ x $k$ and $k$ x $m$, respectively. (Brunet et al., 2004). The value of $k$ is equivalent to the selected factorization rank $p$. Multiple algorithms to calculate NMF exists (Lee and Seung, 2001). Here, we present results from the method described by Lee and Seung, 2001; Brunet et al., 2004), as this created the best fit to the data. The matrices **W** and **H** are randomly initialized, and are updated with the formula given by (Brunet et al., 2004)

$$H_{au} \leftarrow H_{au} \frac{\sum_i W_{ia} X_{iu}/(WH)_{iu}}{\sum_k W_{ka}} \tag{11}$$

and

$$W_{iu} \leftarrow W_{iu} \frac{\sum_u H_{au} X_{iu}/(WH)_{iu}}{\sum_v H_{av}}. \tag{12}$$

The NMF analysis was run in R Statistical Software with the NMF-package (Gaujoux and Seoighe, 2010; R Core Team, 2019). The factors from NMF are labelled as FN in the results.

### 3.1.5    Calculation of the contribution of an ion to a factor/component/cluster

From all these methods two factorization matrices (time series and factor contribution) can be produced at the end. In PMF and NMF, both factorization matrices are calculated simultaneously, whereas in EFA, PCA and PAM the factor/component

time series are calculated after the main algorithm. The factor/component time series show the behaviour of each factor/component during the experiment while the contribution of the different variables to each factor/component (factor/component scores or factor profiles) can be interpreted as the chemical composition of each factor/component. To help the reader visualize the similarities and differences in the results between EFA, PCA, PMF, and NMF in this paper, we calculated the "total factor contribution" of each factor/component to each ion, i.e., how much each factor/component contributes to the signal of a single ion. For PMF and NMF, the values in the factorization matrices (**F** and **H**, respectively) were extracted for each ion and scaled with the sum over all factors for each ion. For EFA and PCA, the absolute values of the loadings were calculated for each ion in each factor/component and then scaled by the sum of all factor loadings. This approach allowed us to compare the division of the ions in each factor/component between the different methods. However, this type of approach conceals the information of the negative factor loadings in EFA and PCA (which are included in the calculation of factor/component time series as weights), but instead visualizes the general contribution of an ion to a factor. Negative factor loadings may have different interpretations. It may indicate that the compound has decreasing effect on the factor, i.e. it acts as a sink for the compounds with positive loading in the same factor. In chamber experiments, negative loading may also refer to decreasing concentration of the compound participating in chemical reactions if it acts as a precursor for other compounds in the same factor. One example of this is benzene, detected (as $C_6H_7^+$) by PTR-MS. When inspecting the original loading values from EFA, for example, it has negative loading in FE1 (identified as later/slowly forming products), and positive loading in FE4 (identified as precursors from car exhaust/background). As benzene originates from the car exhaust, it contributes positively to FE4. However, as it oxidizes over the course of the experiment (thus has decreasing concentration), it has a strong correlation with oxidation products but appears negative in FE1, which mostly includes those later generation products.

## 3.2     Clustering methods

### 3.2.1     Partitioning Around Medoids (PAM)

Partitioning around medoids (PAM) or k-medoids is a clustering algorithm in which the dataset is broken into groups in which the objects or observations share similar properties in a way that object in a cluster are more similar to each other than to the objects in other clusters. The PAM algorithm is fully described elsewhere (Kaufman and Rousseeuw, 1990). Briefly, PAM minimizes the distances between the points and the centre of the cluster (i.e., the medoid) which, in turn, describes the characteristics of the cluster. The distance matrix (often also referred to as dissimilarity matrix) from the observed data can be calculated in many ways. Here, the data was first standardized by subtracting the mean of each ion over the time series and scaling each ion with the standard deviations of the ions. Then, the Euclidean distances (Rencher and Christensen, 2012) were calculated between the ions before providing the distance matrix for PAM. The selection of suitable distance metrics can be challenging and depends on the application and the data. For example, Äijälä et al. (2017) tested four different metrics in their study of  pollution events. In our study, also two other distance metrics were tested; Manhattan distance (e.g., Pandit and

Gupta, 2011) and correlation-based distance metric. The results, however, were similar to those acquired with Euclidean distances, and therefore not shown here. The clustering was performed in R Statistical Software applying the factoextra- and cluster-packages (Kassambara and Mundt, 2017; Maechler et al., 2018; R Core Team, 2019). Clusters are labelled as CL in the results.

Often clustering is applied to cluster the observations in the data (e.g. samples, time points). Here, we have applied the clustering to the variables instead, to group similarly behaving chemical compounds together. This means our calculated distance matrix provides the distances between the variables (i.e., ions), and the centre of the cluster is the "characteristic" ion for that specific cluster. The larger the distances, the "farther apart" the ions are, and ions with shorter distances should be assigned to the same cluster. There are several clustering methods especially meant for clustering of variables (Vigneau, 2016).

The time series for clusters are calculated by summing the concentrations of the compounds in the specific cluster. The interpretation of the results from cluster analysis slightly differs from the interpretation of the results of the other SDRTs. Due to the nature of cluster analysis in general (except fuzzy clustering, see e.g., Kaufman and Rousseeuw, 1990), the variables (here ions) are strictly divided between the clusters whereas for the other SDRTs presented in this study, one ion may have different weighing parameters for different factors/components. Depending on the aim of the study and the type of the data,

this property of cluster analysis may be considered either as an advantage or disadvantage. One obvious advantage of cluster analysis (or hard division techniques in general) is computational time, especially if analysing long ambient data sets. For laboratory measurements, this most likely is not an issue. Hard division techniques have also been shown to work efficiently for VOC measurements when distinguishing between different coffee types (espresso capsules), where strict separation between clusters is needed, as shown in Sánchez-López et al. (2014). For source apportionments studies, where one variable

might originate from multiple sources, cluster analysis using hard division technique is probably not as suitable, as softer division techniques, which can assign one variable to multiple sources/factors.

### 3.3    Determining the number of factors/components/clusters

One of the most difficult tasks in dimension reduction is the choice for the new dimensions of the data. For EFA and PCA, there exist multiple different methods determining the suitable factor and components number. However, these are often more

guidelines than strict rules when handling measurement data as the processes creating the compounds which were measured can be somewhat unpredictable at times. Additionally, as EFA and PCA were originally developed for normally distributed data, tests for determining the number of factors may be influenced if the criterion of normality is not met. Furthermore, the existing tests to investigate multivariate normality are often oversensitive, e.g., for outliers (Korkmaz et al., 2014), which may influence the results. The analysis results from EFA and PCA, however, can be reasonably interpreted despite the data

distribution, as the normality of the data mainly enhances the outcome and is not stated as a strict requirement (Tabachnick and Fidell, 2014). In addition, the two calculation methods selected for EFA were used as they are supposed to be more suitable for non-normally distributed data. ml-EFA is rather insensitive to changes in data distribution (Fuller and Hemmerle, 1966),

whereas pa-EFA is actually suggested to be more efficient if the normality condition is not met (Fabrigar et al., 1999). In this study, the multivariate normality of the data was nonetheless investigated, and results are reported in the SI material.

For EFA and EVD-PCA, we used the scree test first introduced by (Cattel, 1966), Kaiser criterion (Kaiser, 1960) and parallel analysis (Horn, 1965) to investigate the suitable number of factors/components. In the scree test, the factor number is estimated by plotting the acquired eigenvalues (or explained variance) as a function of factor number (see e.g. Fig. 1c). A steep decrease or inflection point indicates the maximum number of usable factors. The Kaiser criterion suggest discharging all factors that have eigenvalues less than 1 (see, e.g., Fig. 1d). In parallel analysis, an artificial data set is created, and the eigenvalues are compared to the eigenvalues of the real data. Here, we created the artificial data set for parallel analysis by resampling the actual measurement data by randomizing across rows as suggested by Ruscio and Roche, 2012). For SVD-PCA, the inflection point can be inspected, e.g., from a plot where the explained variance is plotted as a function of component number (see, e.g., Fig S11 in SI section S3.1). In addition, for EFA, we calculated the standardized root mean residuals (SRMR; Hu and Bentler, 1998) and empirical Bayesian information criteria (BIC; Schwarz, 1978) -values. These metrics measure slightly different properties of the model. BIC is a comparative measure of the fit, balancing between increased likelihood of the model and a penalty term for number of parameters. The SRMR is an absolute measure of fit and is defined as the standardized difference between the observed correlation and the predicted correlation. See S3.2 for more details. A steep decrease in the SRMR values could indicate the number of factors similarly to the scree-test with eigenvalues. From BIC, the minimum value suggests the best fitting model. It should be noted, however, that these methods may suggest slightly different number of factors/components. In addition, many statistical tests are often oversensitive if the data is not completely normally distributed (Ghasemi and Zahediasl, 2012), even if large sample sizes might improve test performance, and therefore, the final decision of the number of factors should be made after evaluating the interpretability of the results.

The suitable number of clusters for PAM was investigated with the Total Within Sum of Squares (TWSS; e.g. Syakur et al., 2018) and Gap statistics (see, e.g., Fig. 1e and 1f). Within-cluster sum of squares is a variability measure for the observations within a cluster, and for compact clusters the values are smaller as the variability within the cluster is smaller. By calculating the TWSS, preliminary guidelines for the number of clusters can be derived by inspecting the inflection point of the TWSS versus number of clusters graph (often referred as the "Elbow method"). In Gap statistics, described in detail, e.g., by Tibshirani et al., 2001), the theoretically most suitable number of clusters is determined from either the maximum value of the statistics or in a way that the smallest number of clusters is selected where the gap statistics is within one standard deviation of the gap statistics of the next cluster number.

Such straightforward statistical tests are not available for PMF, but one possible option is to inspect the relation between $Q$ and $Q_{expected}$. Ideally, the value of $Q_{expected}$ corresponds to the degrees of freedom in the data (Paatero and Tapper, 1993; Paatero et al., 2002) and when $Q/Q_{expected}$ (hereafter $Q/Q_{exp}$) is plotted against the factorization rank, an inflection point may be notable and addition of factors does not significantly change the minimum value of $Q/Q_{exp}$ (Seinfeld and Pandis, 2016). It should be noted, however, that even if the $Q/Q_{exp}$ summed over all ions and time steps is low, the corresponding values of individual ions may still either be rather large or very small, thus compensating each other, and resulting in a unreliably good overall $Q/Q_{exp}$

value (Interactive comment Paatero to Yan, 2016). In addition, the used error scheme in PMF has a large impact on the $Q$-values. If the true measurement error was used, $Q/Q_{exp}$ approaches a value of 1. If the chosen error values were larger than this, the $Q/Q_{exp}$ values will approach a final value smaller than 1. Note that the shape of the $Q/Q_{exp}$ versus number of factors curve is not affected much by the chosen error scheme (see, e.g., Fig. 2b). Therefore, this method should be rather used as a first

suggestion than a strict criterion. A more empirical method for determining the number of factors and interpretation of them exists when investigating ambient AMS data. The acquired factor mass spectra from PMF can be compared to spectra from known sources (Zhang et al., 2011). The time series of these identified factors are then compared to tracer compounds for these factors measured with other instruments (e.g. NOx for traffic emissions, black carbon for burning events). If several factors correlate with the same tracers it is very likely that too many factors have been chosen. An extensive data base of factor spectra

exists for AMS data and it is maintained by the community ([http://cires1.colorado.edu/jimenez-group/TDPBMSsd/](http://cires1.colorado.edu/jimenez-group/TDPBMSsd/), last visit on 9.9.2019). The PMF evaluation tool for Igor Pro used in this study also provides other indexes, including the "explained variance/fraction of the signal", which is shortly discussed in section 3.4.

Several approaches exist for NMF for selecting the factorization rank $p$, but the choice of which method to use is not straightforward (Yan et al., 2019). Brunet et al., 2004) suggested to select the factorization rank based on the decrease in the

cophenetic correlation coefficient (CCC), i.e., at the first value of $p$ where the coefficient decreases (see, e.g., Fig. 2a). In addition, we investigated the cost function that approximates the quality factorization as a function of the factorization rank $p$. For the brunet-algorithm that we applied in this study, this cost function is the divergence between data matrix X and approximation WH (see Eg. (3) in Lee and Seung (2001)).

### 3.4     Determining the "goodness of fit"

When analysing the datasets, we realized that all of the factorization methods in this study are sensitive to even small changes in the data. In order to cross-validate the calculated factorization and approximate the uncertainty in the factors , 20 resamples of the measurement data were created with bootstrap-type sampling (Efron and Tisbshirani, 1986), i.e., sampling with replacement from the original data. The resamples were formed by taking random samples (by row) from the measurement data with replicates allowed while preserving the structure of the time series. The different methods were then applied to the

resamples to validate if the factorization created from the original measurement data was real and the created factorization was robust enough to maintain the achieved factor structure even if minor changes would appear in the data. Simplified, this variation in the factorization for the bootstrap type resamples can be understood as an uncertainty for the factorization results. If we had true replicates of the data set, a similar approach could be used, as in theory the same, repeated experiments with similar chemistry should include the same factors, and the occurring variation in the factorization illustrate the uncertainties

in the factorization.

In addition to the cross-validation of the factorization, the results should be evaluated in a way that we are able to justify how well the factors/components/clusters represent the original data and the underlying information. Often in studies where either EFA or PCA has been used, Explained Variance (EV) is reported for the solution. In principle, the EV could also be used as a

guide when selecting the number of factors by selecting factors until EV reaches "appropriate" value or does not change drastically when more factors are added. In PCA, EV for each component is calculated by dividing the eigenvalue of each component by the sum of the eigenvalues. The sum of EVs for all n-1 components (n is the number of variables in the data) equals 1. In EFA the EV for the factor $k$ (with $p$ factors in total) can be calculated by

$$EV_k = \frac{\Sigma_{i=1}^{n}(\lambda_{ik}^2)}{\Sigma_{i=1}^{n}\left(\Sigma_{j=1}^{p}\left(\lambda_{ij}^2\right)+diag(\mathbf{S-R})\right)}, \qquad (13)$$

where $\lambda_{ij}$ is an element from the loading matrix, $\mathbf{S}$ is the original correlation matrix, and $\mathbf{R}$ is the reconstructed correlation matrix ($\mathbf{R} = \lambda\lambda^{\mathrm{T}}$) (Revelle, 2018). Depending on the algorithm used to calculate EFA, the calculation of EV may vary. In PMF, the calculation of EV is not possible this way, as PMF factorizes the data matrix instead of the correlation matrix. Instead, for PMF there is a possibility to calculate, e.g., the "explained fraction of the signal" for the reconstructed factor model. This can be calculated by comparing the original total time series (sum of the data columns, i.e., individual ion time series) to the reconstructed one by

$$Frac = \frac{mean\left(\Sigma_{j=1}^{n}\left(x_{ij}^*\right)\right)}{mean\left(\Sigma_{j=1}^{n}(x_{ij})\right)}, \qquad (14)$$

where $x_{ij}^*$ is the element from the recalculated data matrix $\mathbf{X^*} = \mathbf{GF}$ (see Eq. (7)), $x_{ij}$ is an element from the original data $\mathbf{X}$, and n is the number of columns (variables, ions) in the data. The disadvantage of this method is the use of the mean. If the signal is both over- and underestimated at different parts of the data, the explained fraction of signal is still very good even if the fit is not. For NMF, a similar index could be calculated. However, due to the differences between EFA/PCA, NMF/PMF, and PAM (uses a fundamentally different approach), the indexes calculated with Eq. (13) and (14) are not comparable between the methods, and therefore not presented here. Instead, we aim for more universal ways to compare the SDRTs.

For NMF and PMF, it is possible to back-calculate how well the created factorization can reproduce the information in the original data. This method is rather straightforward, as both factorization matrices from NMF and PMF are limited to positive values. This allows us to calculate the reconstructed total signal for NMF/PMF, which can be compared to the original total signal to produce residuals. For EFA and PCA, the calculation of the total signal is not possible from the created factorization in a similar fashion, as the acquired loading values (contribution of an ion to a factor/component) may be negative. Therefore, for EFA and PCA the reconstruction is possible only for the correlation matrix, as it is also the matrix that is factorized during the calculation process. This allows us to compare the original correlation matrix to the one produced by EFA or PCA in a similar manner as for the whole data in PMF and NMF. However, due to our large data size, the visualization of the residual correlation matrix is difficult, and instead we have calculated the mean and interquartile ranges (IQR: $Q_3$-$Q_1$) for the absolute values of the residuals. The theoretical minimum value for the mean and IQR is 0, indicating perfect reconstruction, and the theoretical maximum value, i.e., poor reconstruction, is 1. For example, for a variable-pair having a correlation coefficient of 0.7, a mean absolute correlation residual of 0.02 and an IQR of 0.04, this would mean the model over- or underestimates the correlation by 2.86% (= (0.02/0.7) * 100). An IQR of 0.04 would mean that 50 % of all variable-pairs with correlation of 0.7 are within 5.7% (= (0.04/0.7) * 100) of the original value of 0.7.

A very important criterion for the quality of the factorization is the interpretability of the results. If the interpretation of the factors is impossible, the results are useless for the data analysis. Note that all methods presented in this paper are purely based on mathematics, and the "best" result is obtained by solving a computational problem not connected to the real processes in the chamber and instruments leading to the measured dataset. Thus, the user has to apply the available external information (e.g., about possible reaction products or if ions should be split between multiple factors) to validate the feasibility of a factorization result. But it is a fine line between applying this prior knowledge about the possible chemical and physical processes in the chamber to validate a factorization result and dismissing an unexpected feature discovered by the factorization method as "unphysical" and thus wrong. Applying more than one factorization method may be helpful to protect the user from dismissing unexpected results.

## 4    Results and Discussion

### 4.1    Gas phase composition from PTR-MS

#### 4.1.1    EFA

Figure 1 shows results for the tests described in section 0. The eigenvalues and parallel analysis results for EFA are not shown, as the results were very similar to those acquired for PCA. Also, the factorization results from ml-EFA and pa-EFA were so similar, that only the results from ml-EFA are presented here. Fig. 1a shows the empirical BIC and Fig. 1b the SRMR values for factorization ranks ranging from 1 to 10. The minimum value in empirical BIC was achieved with 4 factors and the inflection point in SRMR also lies around 4 factors.

As all these tests suggest a 4-factor solution for PCA and EFA, we compared the factor time series and factor contribution for the 4-factor (Fig. 3) and 5-factor solution (Fig S9 in SI section S4.1) for EFA with Oblimin rotation. The additional FE5 seems to be a mixed factor with small concentration created from FE4 and FE2, instead of a new factor with different properties. The original loading values for the 4-factor solution are presented in Fig. S10 as a scatter plot.

The variation in the factors from the resamples is largest around the start of the photo-oxidation, as expected, when there are fast and large changes in the concentrations. The mean and IQR for the absolute values of residual correlations for the 4-factor solutions were 0.0109 and 0.0108, indicating good reconstruction.

In the following, we interpret the factors for all SDTR methods based on their characteristic factor time series shape and the identified compounds in the factors. An overview of this interpretation is given in Table 1. Based on the shapes of the factor time series, FE1 can be identified as an oxidation reaction product factor. It starts increasing slowly when the photo-oxidation starts, so these are either products from slow reactions, or multiple reactions steps are needed before these compounds are formed. FE2 is also an oxidation reaction product factor, but these are first (or early) generation products which rise quickly after photo-oxidation starts and are slowly removed by consecutive reactions as the photo-oxidation continues and/or by partitioning to the particle phase or chamber walls. FE3 is a precursor factor which shoots up during the α-pinene addition

(slightly after t = -50 min) and is stable until the start of the photo-oxidation. Together with the factor mass spectrum which is dominated by signals at m/z 137 and 81, this is a clear indication that FE3 represents α-pinene in the chamber. Note that although proton transfer is a relatively soft ionisation technique, a certain amount of fragmentation of the mother molecule α-pinene (m/z 137) is observed showing fragments at, e.g., m/z 81 (Kari et al., 2018). FE4 seems to include some car exhaust VOCs and residue from the background. It has very low concentrations compared to the other factors. It decreases slightly throughout the whole experiment and seems not affected by the onset of photo-oxidation.

### 4.1.2    PCA

Figure 1c shows the eigenvalues as a function of component number for EVD-PCA with the results from parallel analysis. In Fig. 1d the eigenvalues for the first two components are omitted to show the changes with more components better. The blue line shows the Kaiser criterion (eigenvalue = 1). SVD-PCA (when applied to scaled data matrix) was not able to separate α-pinene as its own component, but instead created two factors which were dominated by the unreacted α-pinene and its fragments (see Fig. S12 in SI section S4.1). In addition, the unrotated solution included a large number of negative loadings, which complicated the interpretation of the components. No improvement was achieved when SVD-PCA was applied to the data matrix without any scaling (see. Fig. S13). Oblimin rotation was applied to create factors that could be interpreted in a physically more meaningful way, but the algorithm did not converge. So this is a case where the result of the factorization method is very difficult to interpret or even contrary to the available information (e.g. the α-pinene precursor behaviour). As additionally the underlying algorithm is struggling with the dataset (i.e., not converging), we will not discuss these results in detail here but rather focus on the EVD-PCA.

The number of components indicated by parallel analysis is 4 (Fig. 1c), but the eigenvalues decrease below 1 only with 10 components (Fig. 1d), indicating 9 components should be selected. However, the eigenvalues for the components 5-9 are rather close to the Kaiser limit (between 1.47 and 1.04, respectively), and therefore the 4-component solution was selected. In addition, the "knee" in the eigenvalues is around 4 or 5 components, but as for EFA, the addition of a fifth component did not create a new component with different properties but mixed properties of the previous components.

Figure 4 shows the component time series and total contribution from EVD-PCA with Oblimin rotation and the original loading values for the 4-component solution are presented in Fig. S14 as a scatter plot. Oblique rotation was used despite the orthogonality assumption of the components, as for true physical components the assumption of orthogonality is not that realistic either, as it would indicate that the chemical processes taking place in the chamber don't have any correlation between the different processes. Oblique rotations allow correlation between the components, meaning the detected ions in different components interact with each other. For example, the decrease of α-pinene concentration is mostly caused by chemical processes which in turn form other ions detected by PTR-MS. Additionally, there are multiple consecutive processes

(reactions) at work simultaneously, so the correlation between the components is not a straightforward indicator of connected processes, but it is more realistic than no correlation at all.

The mean and IQR for the absolute values of the residuals of the correlations were 0.0116 and 0.0107, respectively. Compared to the EFA solution with 4 factors, the residuals are slightly larger. The total contribution of compounds to each factor is very
similar for EFA and EVD-PCA (Figs. 3b and 4b, or Figs. S10 and S14), which agrees with the very similar factor/component time series in general in the Figs. 3a and 4a. The interpretation of the components CO1, CO2, CO3, and CO4 is therefore the same as above for the EFA factors FE1, FE2, FE3, and FE4, respectively.

### 4.1.3    PAM

The test parameters TWSS and gap statistics for PAM are shown in Fig. 1e and 1f. The TWSS vs number of cluster values do
not show a clear inflection point, but it could be roughly assigned between 3 and 5 clusters. There is no maximum value reached with gap statistics which indicates the theoretical number of clusters is 9, as there the gap statistic is within one standard deviation of the gap value in the 10-cluster solution. However, the increase of the gap value clearly slows down after 3 clusters. After careful evaluation, the 4-cluster solution is determined as most interpretable, and the cluster time series and distribution of the ions are shown in Fig. 5. Four clusters were selected, as the selection of only 3 clusters (Fig. S15 in SI
section S4.1) is not enough to explain the variation in the data, as the addition of one cluster reveals new features. On the other hand, the 5-cluster (Fig. S16 in SI section S3.1) solution seems to split off an additional low-concentration cluster from CL4 (Fig. 5a), instead of showing a new distinct cluster. The distinction between the "more correct" solution with 4- or 5-clusters is not, however, straightforward, as CL5 (Fig. S16) could be interpreted as a car exhaust precursor cluster, as is CL4 in Fig. 5. Clustering statistics are presented in Table S2 (SI Sect. S4.1).
When comparing the shape of the cluster time series to the EFA and PCA results in Fig. 3 and 4, the results agree well. The largest difference appears in the CL4, which has larger concentrations in PAM compared to FE4 acquired from EFA and CO4 from PCA. The shapes of FE4 and CL4 are also slightly different, as CL4 has a small decrease in the concentration at = 0 min whereas FE4 is barely affected. However, comparing the actual concentrations between these methods (EFA/PCA and PAM) may be misleading, as in EFA and PCA, the acquired loading values are used as weights when calculating the factor time
series whereas in PAM the time series of the clusters are calculated as a direct sum of the cluster compounds, as explained in chapter 3.2.1.

Dichotomized loadings (for each ion: 1 for factor with largest loading, 0 for the other factors) for EFA were tested to see if then the results agree better with those from PAM as in PAM there are no loading values, meaning an ion either is in a cluster or not. With dichotomized EFA loadings we make the same assumption: one ion is classified to one factor only, and therefore
stemming from only one source or source process. Figure S17 (SI Sect. S4.1) shows the results from dichotomized EFA. When compared to PAM (Fig. 5), the factor/cluster concentrations agree well, but there are clear differences in the ion distribution. EFA classifies the weak ions with low concentration to the product factors (FE1, FE2), whereas PAM assigns them to the background/precursors cluster (CL4).

### 4.1.4   NMF

Figure 2a shows the divergence of the cost function D(X||WH) and CCC for factorization ranks from 2 to 10 for NMF. The CCC has a first decrease in the values at the rank 4 and the D(X||WH) shows an inflection point around the ranks 4-5. Figure 6 shows the factor time series and total contribution for the NMF with factorization rank 5. Five factors were selected, even though CCC suggest only 4 factors, as the addition of one factor to the 4-factor solution (Fig. S18 in Sect. S4.2) did add a new feature to the solution in contrast to the SDRTs presented above. FN2 in the 4-factor solution decreases drastically between t = -50 and t = 0, indicating it might include background ions, but on the other hand, it also peaks right after t = 0, indicating oxidation products also contribute to that factor. These mixed properties in the factor FN2 indicate more factors are needed and indeed in the 5-factor solution this contradictory behaviour no longer occurs.

Similar to the results shown above, the range in the factor time series for the bootstrap replicates is larger when the factors exhibit fast changes in the concentration (Fig. 6a). In addition, FN3 from the real measurement data has a lower maximum concentration when compared to the bootstrap replicates. This indicates NMF is rather sensitive to the small changes in the data and only a few deviant observations present in data but not in majority of the resamples can cause this kind of discrepancy. Factors FN1-4 seem to correspond to the same factors found with EFA, PCA, and PAM (Fig. 2, 3 and 4), and especially α-pinene is clearly assigned to the same factor (FE3/CO3/CL3/FN3) in all the used methods. FN5 in NMF, however, has properties that were not detected (or separated from others) with EFA, PCA, or PAM even if more factors were added. This new factor could be interpreted also as oxidation product factor, but as it increases slower and decreases later than the early product factor (FN2), it mostly includes intermediate products. These are most likely compounds which are formed through (multiple) reactions and consumed in further oxidation reactions.

By recalculating the data matrix, **X**, with the original factorization matrices **W** and **H**, we can inspect how well it has been reproduced. Here, the total signal (total time series) is then calculated by summing all ions for each time step for the original data matrix and the reconstructed data matrix. The differences between the original total signal and the one produced by NMF (i.e., the residuals) were smaller than $10^{-10}$, indicating a good mathematical reconstruction. The boxplot of the residuals with 4 and 5 factors is shown in Fig. S19 (SI Sect. S4.2)

### 4.1.5   PMF

The acquired $Q/Q_{exp}$ for different factorization ranks in PMF with the constant error scheme are presented in Fig. 2b. The values are the minimum values from all possible solutions with fpeak values from -1 to 1 by a step of 0.5. The $Q/Q_{exp}$ were at the minimum at fpeak = 0 with the number of factors (1-10) tested. We notice that the values are slightly smaller in general when using the signal following error as the absolute values of the errors in this error scheme are significantly larger around fast changes than in the static error scheme and thus decreasing the observed $Q$ values (see Fig. S4 for the different error schemes). Values for the signal following error decrease slightly below 1 (0.88) for the 5-factor solution, whereas with the static error they stay above 1.91. After careful evaluation of the results with different number of factors, the solution with 5

factors (Fig. 7, fpeak = 0) was selected to be presented and interpreted here. The solutions with fewer factors were inconclusive, and the addition of a fifth factor did add a new feature. The results with 4 factors are shown in the SI material (Sect. S4.2, Fig. S20). In the 5-factor solution, the solid lines in the time series are the results for the measured data, and the shaded areas show the ranges for the bootstrap resamples.

For the static error case, the factorization from resamples agree well with the ones from measurement data. For the signal following error (Fig. 7c-d) the differences are significantly larger, and for example FP5 has a larger peak concentration than in any of the resamples. This is most likely caused by few deviant values in the data which are not present in the resamples, and thus creating smaller peak concentration for FP5 for the resampled data. Resampled data includes more sudden changes due to added and/or missing data rows, thus causing PMF to perform poorer. In addition, the other variations in the resampled

ion time series may cause ion contributions (especially those originally assigned in-between factors)  to shift slightly from FP5 to FP2, as for FP2 the values in the time series are higher in the resamples compared to the original data. This difference between the error schemes is caused by the error values themselves. For the signal following error, the factorization is more "precise" (less wiggly factors), but even small shifts in the data (bootstrap resamples) distort the factorization more than in the static error case.

When comparing the results for the measurement data with different error schemes in Fig. 7, we note that the α-pinene precursor factor is slightly less pronounced with the signal following error, i.e., the solving algorithm assumes these fast changes are not "real" but rather outliers. This is caused by the used error scheme, where errors are larger for the fast changes in the data (Fig. S4b). In ambient data not measured at instant proximity of strong emission sources, for which PMF is often used, this type of error is beneficial as there the fast changes are more likely to be noise or instrument malfunctions (excluding,

for example, sudden primary emission plumes), and we are more interested of the long-term changes instead. For laboratory data, where large changes are often caused by rapid changes in actual experimental conditions, e.g. due to injecting α-pinene or turning the UV lights on, the static type of error is most likely preferable. Usage of the static error scheme helps to avoid overcorrecting intentional (large) changes in experimental conditions and confusing them with real variation taking place during the experiment and typically being much less pronounced.

Figure 8 shows how well the original data matrix can be reproduced with the created factorization matrices. The residuals for the static error are generally larger as most of them are in the range $0 \pm 0.5$ (for signal following error $0 \pm 0.15$), but there are much larger "outlier" values for the signal following error. This is due to the structure of the signal following error, which is larger during the fast changes in the data, as shown in Fig. S4b (SI Sect. S1.4). For the static error the residuals vary more throughout the whole data, whereas for the signal following error the residuals are smaller, but a few rather large values appear

at the start of the photo-oxidation, as seen in Fig. 8b. This highlights the role of the selected error values in PMF, which act as weights for the data. Smaller error value means that the corresponding $Q$ value at this time will be much larger and an improvement of the model at this part of the data will have a big impact on the optimisation value. This means that the error values can be used to emphasise certain parts of the dataset which otherwise would not be recovered very well by PMF. Note that this is a key difference to NMF where no error-based weighting of the data is done.

#### 4.1.6 Comparing the SDRTs applied to gas phase composition

Table 1 summarizes the acquired results from different SDRTs for the gas phase composition data measured with PTR-MS, and Figs. S21 and S22 in SI section S5 show separate factor contributions for each of the SDRTs. Comparison of the total factor contribution for some selected compounds for the 4 factors from EFA, PCA and PAM are shown in Fig. 9. We note that the differences are very minor between EFA and PCA, and hardly visible in the coloured bars. When compared to PAM, we see that e.g. acetaldehyde and methyl ketene are assigned to the red cluster (CL2), which is also dominating in EFA and PCA (FE2, CO2). Figure 10 shows the same compounds for NMF and PMF. There, the largest difference is between the 2 oxidation product factors, coloured as black (late oxidation products, factor1 in Table 1) and red (fast oxidation products, factor2 in Table 1). For the selected compounds, NMF has more weight assigned to the fast forming products than PMF. In addition, PMF assigns much more weight to the intermediate oxidation product factor (pink) for some of the compounds.

The factorization acquired from PMF agrees well with the factorization from NMF when comparing the factor time series, as expected since the methods are rather similar. Comparison of the concentrations of the factors between PMF and NMF directly is not exact, as these methods have different weighting between the produced factorization matrices due to the different solving algorithms. The largest difference is the early product factor, FN/FP 2. In NMF (Fig. 6), this factor (FN2) increases from 0 ppb to 30 ppb very fast at t = 0 min, then it decreases rapidly to just above 20 ppb and continues to decrease almost linearly towards 10 ppb. In PMF, this factor (FP2, Fig. 7) has a similar increase at t = 0 min, but it decreases exponentially instead of the fast drop and constant decrease present in NMF solution. The different slope has direct implications for the interpretation of the factor. A faster decrease is interpreted as a faster removal/destruction process for ions classified into this factor. This is typically related to reaction speeds or to how far along a product is in the chain of oxidation reactions. When comparing the total contribution of FN2 in NMF and FP2 in PMF, in NMF ions with m/z 90-100 have much more contribution to FN2, whereas in PMF these ions seem to be assigned to FP1 instead. Otherwise the factors agree well with those acquired from NMF, and their interpretation is therefore similar: 3 oxidation product factors (FP1, FP2, and FP5), 1 background/car exhaust precursor factor (FP4), and 1 α-pinene injection factor (FP3).

Another important difference between NMF and PMF is the relation between the factors at the end of the experiment. In PMF, at the end everything is shifted to FP1 (later generation oxidation products) and the other factors decrease to 0, whereas in NMF there still is contribution from the other oxidation product factors FN2 and FN5 in addition to FN1. More fundamental studying of the algorithms for both PMF and NMF is needed to explain this behaviour.

The factorization acquired with EFA, PCA and PAM is more robust compared to NMF and PMF when inspecting the bootstrap ranges in the top panels in Figs. 3, 4, 5, 6 and 7. This may be explained with the different number of factors (4 or 5) as with more factors, one factor includes less (strongly) contributing ions, which causes factorization to vary more when the data is different. But most of the differences between these SDRTs are still explained by the methods themselves and the solving algorithms. PMF and NMF are more sensitive to small changes in the data, whereas EFA, PCA and PAM succeed more reproducibly in finding larger structures and changes in the data.

Addition of a fifth factor to EFA, PCA and PAM did not add a factor showing a new feature, as it did in NMF and PMF, but a sub-factor. This sub-factor has very low concentration, but if inspected separately (not shown), it peaks around t = 0 min similar to the factor2. This means, instead of adding a factor consisting of intermediate oxidation products (as in NMF and PMF), the added factor is another early product (or background) factor. This is also caused by the difference in the methods, as these 3 SDTRs (EFA, PCA, and PAM) concentrate more on the fast changes (which take place here at t = 0 min), whereas NMF and PMF focus more on slow changes. This is one example where the chosen method (EFA/PCA or NMF/PMF) has a direct impact on the interpretation of the data. For understanding the chemical processes in the experiment, the existence of 2 or 3 oxidation product factors is of great importance.

Factor4 has different behaviour in the time series in EFA and PCA compared to NMF, PMF and PAM. In the latter SDRTs this factor starts decreasing immediately and the concentration drops through the whole experiment, implying it is affected by car exhaust precursors that are oxidized, and the products are assigned to other factors later on. In EFA and PCA, this factor has small and rather stable concentration over the time series (also in addition to a small contribution to the total signal), suggesting it could consist of background compounds present throughout the whole experiment. Without exact identification of all the compounds present in these factors (which is out of the scope of this study) , it is hard to say if this difference is real or if it is related to the different calculation of the SDRTs. We provide more details of the comparison between the factors from different SDRTs in Sect. S5 in SI material.

## 4.2    Particle phase composition from AMS

### 4.2.1    EFA and PCA

As the AMS data from the experiment includes only one observation about every 10 minutes, the data has many more variables (compounds) than observations. This causes problems for EFA and EVD-PCA, as those methods are based on the correlation matrix, which will not be positive definite due to the small number of observations (rows) compared to the number of variables (columns). In EVD-PCA, the second step is to calculate the eigenvalues which in this case may also be negative and result in a non-interpretable outcome. With this type of data, the results of EFA are also sensitive to the used algorithm. The calculation in ml-EFA did not converge at all, but pa-EFA was able to produce results. Due to these restrictions in the calculation process, the results from EFA and PCA are only briefly discussed below, and example figures can be found in the SI material (Sect. S6.1). In addition, due to the very small data size, the bootstrap-type resampling of the data has a too drastic effect on the data structure to validate the repeatability of the factorization and is therefore not applied for any of the SDRTs.

Figure S23a and b (SI material Sect. S6.1) shows the results for the tests investigating the correct number of factors and components pa-EFA and EVD-PCA. For EFA in Fig. S23a, the empirical BIC reaches a minimum value with 4 factors and the inflection point in SRMR is at 4 factors. For EVD-PCA, however, parallel analysis (Fig. S23c) suggests only 1 component, mainly indicating the data is not suitable for PCA at all. The eigenvalues also do not reach 1 (Kaiser criterion) with up to 10 components tested. The eigenvalues for EFA (not shown) reached 1 for the 6-factor solution, and parallel analysis results (not

shown) indicated to select only 1 factor. The differences in these test results is mostly caused by the computational issues mentioned above. Indeed, neither EFA nor PCA (SVD or EVD) were able to separate more than two factors/components from the data, when 2-5 factors/components were tested (see e.g. Figs. S24 and S25 in Sect. 6.1). While a 2-factor solution could be correct in principle, it seems unlikely for the investigated system. The particle phase is constantly formed by low volatile gaseous compounds condensing. As shown above, the gas phase composition changes constantly as compounds are produced and consumed. Thus, it is highly unlikely that during the 4 h of chemical reactions in the chamber the same mix of low volatile compounds is present and condenses onto the particles.

### 4.2.2   PAM

No clear inflection point is visible in the TWSS-plot in Fig. S23e; the value decreases when increasing the cluster number with a small "bump" at 4 clusters. The gab statistics (Fig. S23f) does not reach a maximum value, and it also does not reach the other criteria explained in Sect. 3.3. The inconclusiveness of the tests results may be caused by different reasons, and to investigate this further, PAM was conducted with 2 – 5 clusters and the results are shown in the SI material (Figs. S26 and S27). Increasing the number of clusters from 3 (Fig, S26b) upwards adds clusters with extremely small concentrations and a similar time series shape to the previously found clusters. The very similar shape of the time series of the clusters suggests that only one type of SOA particles was formed quickly after the start of photo-oxidation, and that the chemical composition changed only marginally. Again, this seems unlikely for the investigated system.

The inability of PAM to identify multiple SOA particle types most likely lies in the method itself. Each variable (ion) is assigned to one cluster and cannot be spread over multiple clusters. However, it is well known that AMS applies a "hard" ionisation technique. Thus, a high degree of fragmentation is expected and indeed, most carboxylic acids, for example, are detected as $CO_2^+$ (m/z 44). This means, a highly oxidised organic acid, formed late in the experiment after multiple steps of oxidation will be detected at the same variable (ion) as a different acid formed much earlier. Due to the "one variable-one cluster" method, PAM is incapable of resolving this information in the data. While EFA and PCA could still be used if the data matrix is suitable (i.e., more rows than columns), PAM is unsuitable for this AMS dataset or generally datasets where variables have strong contributions from more than one source.

### 4.2.3   NMF

The D(X‖WH) has an inflection point at factorization rank 4 and CCC shows the first decrease in the values with 4 factors, as shown in Fig. 11a. We selected the 4-factor solution for the detailed interpretation. We discuss additional reasons why the 4-factor solution should be selected in SI Sect. S6.2. The factor time series are shown in Fig. 12a. Fig. 12b shows the original ion-to-factor contributions from NMF without any scaling. The total factor contribution plots are omitted, as we don't have PCA/EFA results to compare. The delay in the time series after t = 0 (before the factors starts increasing/decreasing) is most likely caused by the small time resolution (10 min) of the data. The residuals were on the same order of magnitude as for the PTR-MS data, indicating again very good reconstruction of the original signal.

The mass spectrum of FN1 is dominated by the $C_nH_{2n+1}^+$ and $C_nH_{2n-1}^+$ ion series, conforming to the typical features of combustion-related primary organic aerosol and thus it is interpreted as a hydrocarbon organic aerosol (HOA) -factor. FN1 was originated from car exhaust, as it already appears before t = 0 min. FN1 increases slightly at the start of the photooxidation. The increase is partly attributed to the new formation of the HOA component when the HOA type compounds in the hot

exhaust gas was introduced to the chamber and contains marker ions associated with HOA (e.g. m/z 57, Zhang et al., 2005) condensed again in a cooler chamber. Meanwhile, we can't rule out the possibility that HOA has been produced as a minor product after the photooxidation reaction was enabled in this study. FN2 can be interpreted as α-pinene secondary organic aerosol (SOA) derived semi-volatile oxygenated organic aerosol (αP-SOA-SVOOA) after we carefully compared the factor mass spectra with pure α-pinene experiments conducted at similar settings reported by Kari et al., 2019b). In addition, FN2 is

characterized by the prominent peak as m/z 43. The mass spectra of FN2 and FN4 are rather similar, but FN4 has more contribution from m/z 44, a marker of oxygenated organic aerosol (Zhang et al., 2005) and thus it is identified as an αP-SOA-LVOOA factor. The FN3 was appointed as a mixed LVOOA. Except for the high peak at m/z 44 in the FN3 mass spectrum, its time series is also consistent with the SOA formation in the mixed a-pinene/car exhaust SOA experiments conducted in similar settings (Kari et al., 2019b) and thus it is identified as a mixed LVOOA factor stemming from later generation oxidation

products. Summary of the generated factors from NMF can be found in Table 2.

### 4.2.4    PMF

The $Q/Q_{exp}$ for the two error schemes are shown in Fig. 11b. Neither of the error schemes show a clear inflection point. Examples of behaviour of the errors as a time series are shown in Fig. S7 (SI Sect. S2.2). With the standard AMS error, the $Q/Q_{exp}$ values do not reach 1 (with 10 factors $Q/Q_{exp}$ = 1.76) whereas with the static error the values decrease below 1 for 7

factors. The solutions with 2-5 factors were inspected, and the 2-factor solution (Fig. 13) is presented here as the most interpretable one (summarized in Table 2). The primary OA factor, separated by NMF (Fig. 12, FN2) was only found if using 4 factors and the static error scheme in PMF (see SI Sect. S6.2, Fig. S31a-b). However, interpretation of the time series for that solution was found to be very difficult due to the extreme anticorrelation between the time series, and thus the 2-factor solution was selected. The two factors were interpreted as SVOOA and LVOOA. In addition, the largest relative decrease in

the $Q/Q_{exp}$ was observed with the 2-factor solution.

The residuals for the standard AMS error were smaller, as shown in Fig. 14. This agrees with the analysis of the gas phase data set, where the residual for the signal following error (which has a similar profile in time as the standard AMS error) were generally smaller compared to the static error.

The signal following error, used for PTR-MS, was also tested for particle phase data. However, as this type of error showed

very similar behaviour as a time series as the standard AMS error, and produced very similar outcome, those results are omitted from this manuscript.

### 4.2.5    Comparing the SDTRs applied to particle phase composition

Table 2 summarizes the acquired results from NMF and PMF for the particle phase composition data measured with AMS. PAM was not able to separate distinct clusters due to the inability of clustering techniques to classify an ion into multiple clusters. Comparing the relative factor spectra and fraction of signal from NMF and PMF with the static error (Figs. 12b and 13b), the distribution of ions is similar between the LVOOA in the PMF solution and the mixed LVOOA factor in the NMF solution, and also similar between the SVOOA in the PMF solution and the αP-SOA factor (integrated- αP-SOA-SVOOA and αP-SOA-LVOOA factors). When inspecting the individual ion time series in the original AMS data, most of them have rather "smooth" behaviour, similar to the factors acquired from NMF. It seems, that PMF gives more weight to the background ions (with very small concentration) which do not have that clear structure in their time series, thus including more of their behaviour in the final factors, if the number of factors in PMF is increased from 2 (see SI Sect. S6.2 Figs. S30-S32). Residuals from NMF reconstruction (with 4 factors) were over ten orders of magnitudes smaller (for NMF between $-1.3 \cdot 10^{-13}$ and $7.1 \cdot 10^{-14}$) than those from PMF (between -0.06 and 0.13 for the 2-factor solution with standard AMS error, see Fig. 14), indicating better reconstruction of the data with NMF. Most likely PMF struggles with the small data set, thus not being able to recover all the factors found by NMF and construct reasonable time series for those factors (see 4-factor solution in Fig. S31), whereas NMF does not seem to be affected by the data size. In addition, the weighing between the factorization matrices between NMF and PMF is different not only due to the error matrix that is given as a weight in PMF, but also because of the different solving algorithms for each method. This, on the other hand, assigns different emphasis between the matrices, possibly causing NMF to use more effort to reconstruct the data matrix with factor time series. However, the reader should keep in mind that for detailed chemical analysis of such data set, especially with PMF, downweighing is advisable. In addition, the replacement of very small negative numbers with very small positive numbers is not mandatory for PMF, as it can run with a few negative values into some extent. However, we did the replacement here, as the NMF algorithm used here, does require strictly positive input data. Acquiring a balance between statistically good results and realistic factors might be challenging, and to achieve more robust results, testing different error schemes may be beneficial, especially for a data set with such small size.

### 4.3    Computational cost

To approximate the differences in the computational time between the different SDRTs, the methods were applied with 2-10 factors each, having 9 runs in total for each method. No rotations were applied (no rotation for EFA and PCA, Fpeak = 0 for PMF), as the rotational methods between EFA/PCA and PMF are not directly comparable. Computation times include the calculation of the correlation matrix when needed and calculation of the factor time series for PAM, EFA and PCA (which is calculated outside the main algorithms), as described in Sect. 3.1.1. Three data sets with different sizes were tested, and the results are presented in Table 3. AMS includes the particle phase measurement data (size 26x306) presented in Sect. 4.2. and PTR-MS the gas phase composition data (size 300x133), which was analyzed in detail in Sect. 4.1.PTR-MS*5 is a larger data set created from the gas phase composition data by duplicating the data rows 5 times (final size 1500x133). The computational

times for NMF and PMF were clearly longer when comparing to the other SDRTs. This is not surprising, as PMF and NMF calculate both factorization matrices at the same time, whereas for the other SDRTs only the matrix presenting the contribution of ion to factor is found at first, and the time series of the factors/components/clusters are calculated afterwards. In addition, the PMF2 algorithm used through the PMF evaluation tool for Igor Pro does read and write text files for each PMF run, thus

significantly increasing the computational time.

## 4.4    Summary of the SDRTs used in this study

The methods tested in this study have many similarities and many, fundamental or computational, differences between them. Though in literature, they are many times applied to similar problems. In this section we will summarize some of these properties.

EFA has a fundamental difference to the other methods as it is by definition a measurement model of a latent variable, i.e. the factor, (Osborne, 2014) whereas the other methods are basically describing the measured data with linear combinations of measured variables. The latent variables in EFA, i.e. the factors, cannot be directly measured, but instead, they are seen through the relationships they initiate in a set of Y variables, which are measured. In the other methods, in turn, the factors, components or clusters are calculated directly from the measured variables Y (Rencher and Christensen, 2012; Osborne, 2014).

The approach to data reduction in PCA is to create one or more summarizing variables from a larger set of measured variables by retaining as much as possible of the variation present in the original data set (e.g. Jolliffe, 2002). This is done by using a linear combination of a set of variables. The created summarizing variables are called components. The main idea of the PCA is to figure out how to optimize this process: the optimal number of components, the optimal choice of measured variables for each component, and the optimal weights when calculating the component scores.

The objective of cluster analysis is to divide the observations into homogeneous and distinct groups (Rencher and Christensen, 2012). Cluster analysis is a method where the aim is to discover unknown groups in the data, which are not known in advance. The goal of the clustering algorithm is to partition the observations into homogeneous groups by using some measure of similarity (or dissimilarity), such that the within-group similarities are large compared to the between-group similarities. The choice of the similarity measure can have a large effect on the result. One property of cluster analysis is that it will always

calculate clusters, even if there is no strong similarity present between the variables in the data (Wu, 2012). This should be noted when interpreting the results, especially if the user has no a priori information about the number of clusters.

NMF and PMF provide an alternative approach to the decomposition assuming that the data and the components are non-negative (Paatero and Tapper, 1994; Lee and Seung, 1999). Thus, all the features learned via NMF and PMF are additive; that is, they are adding together strictly positive features. PCA and EFA tend to group both positively correlated and negatively

correlated components together, as they are only looking for the correlations of variables (except SVD-PCA, which can be applied to the data matrix directly). On the other hand, NMF and PMF, by constricting W and H to positive values, find patterns with the same direction of correlation. Thus, NMF and PMF work well for modelling non-negative data with positive correlations. However, if the interest is not only in the positive effects, then PCA and EFA can provide more information of

for the investigated system. Cluster analysis is suitable for classifying observations based on certain criteria. The researcher can measure certain aspects of a group and divide them into specific categories using cluster analysis. However, this method is not suitable for data with variables which should show contributions from multiple factors/components (e.g. strongly fragmented signals in AMS data).

Factorization methods, including those used in this paper, operate on the fundamental assumption that the factor profiles (here factor mass spectra) are constant over the investigated period. Often, this has been interpreted in a way that chemical processes occurring in a chamber experiment or the atmosphere violate this assumption. However, this interpretation is based on a too narrow definition of what a factor represents. A factor can be seen as a direct (emission) source of compounds which changes its contribution to the whole signal (e.g. primary emissions from biomass burning as a fire develops and then dies). But a factor

can also be interpreted as a group of compounds showing the same temporal behavior. If this group is released together as an emission or if the compounds are formed in the same ratio by some chemical process should not matter. In the latter case, it is important how wide the group is selected, i.e. if we group products of processes together for which the contribution changes with time. This means, choosing the optimal number of factors becomes even more important when chemical processes are occurring. EFA and PCA account for the chemistry happening in chamber measurements with negative loadings, as described

above. Same factor can contain educts and products of a chemical process (e.g. oxidation), with the difference that their loadings are negative and positive, respectively.

Taking account of everything above, the most important thing to consider when selecting the SDRT is the interpretation. What are the features the researcher wants to deduct from the data, what are the properties of the data and how the data can answer the research questions. As we have shown in the results section, the methods provide quite similar reconstructions of the time

series, but the interpretation of the steps leading to these are quite different. For example, comparing EFA factor FE4 in Fig. 3a and PMF factor FP4 in Fig. 7a reconstructions, they seem to show the same procedure but the first one includes both positive and negative effects and the second one consists only of positive effects.

## 5   Conclusions

The main objectives in this study were to a) investigate how different SDRTs perform for gas- and particle-phase composition

data measured with mass spectrometers, b) how the interpretation of the factors changes depending on which SDRTs has been used and c) how well the SDTRs were able to resolve and classify the factors representing chemistry behind the investigated data set of photo-oxidation of car exhaust combined with α-pinene. We showed that EFA, PCA and PAM were able to identify 4 factors from the gas phase composition data, whereas NMF and PMF succeeded to separate one additional oxidation product factor. The behaviour of the factors as time series were similar, when considering the differences in the calculation of the factor

time series matrix in different SDRTs. For example, the EFA and PCA factors were nearly identical, and the differences in the interpretation lays more in the definition; principal components are defined as linear combinations of the variables (ions) whereas in EFA the variables are expressed as linear combinations of the acquired factors. From the particle phase data, NMF

was able to separate four factors, whereas PMF separated two. PAM was not able to find more than two separate clusters, most likely due to the high degree of fragmentation in the data and the constrain of PAM to assign one ion to only one cluster, as discussed in Sect. 4.2.2. EFA and PCA had computational constraints due to the small data size acquired from the AMS and could not to be applied. In addition, PMF also faced assumedly computational issues with the small particle phase data set, thus not being able to reasonably separate the HOA factor.

The difference, which might be an advantage or disadvantage depending on the application, of PCA and EFA over PMF and NMF is their use of the correlations of the variables instead of the raw data. When using the raw data, ions with high concentration may dominate and hide interesting behaviour occurring in the lower-concentration ions and instead classify those as insignificant background. When using correlations, the concentrations of the ions do not affect the created factorization until the factor time series are calculated, and in principle, variables with different units can be factorized simultaneously. On the other hand, it may diminish some of the more minor and subtle changes. As NMF and PMF do not rely on the correlations, they are more sensitive to the smaller changes taking place in the data. The disadvantage of signals with high intensity dominating in the analysis can be tackled in PMF by choosing an appropriate error matrix that weights the ion signals. Selection of the error matrix can also be crucial when interpreting the PMF output, as a sub-optimal choice may hide the identification of important properties of the data.

The gas-phase data results from PAM agreed moderately with those from EFA and PCA, when taking into account the ability of PAM to assign one ion to only one cluster instead of multiple ones. When comparing the performance of the SDRTs to the bootstrap-type resampled data, we noted that the factorizations from EFA, PCA and PAM were more robust compared to the PMF and NMF results. Results from PMF with different error schemes were similar, but the static error provided more robust solutions when applied to the bootstrap-type resamples.

The findings by Koss et al., (2020) proposed that HCA can be used to quickly identify major patters in mass spectra data sets, which is in agreements also with our results from PAM. Our findings for PMF partly differ, as they suggest that PMF is not able to sort chemical species into clear generations by their oxidation state. In our study, we found 3 factors (factors 1, 2 and 5, see Table 1), which can be interpreted as representatives for different oxidation states. However, they can also present reactions taking place with different reaction kinetics (faster and slower reactions), as discussed in the results. In addition, Koss et al., (2020) used gas-phase data from $I^-$-CIMS and PTR3 with $NH_4^+$ as a reagent ion, which are more sensitive to later generation oxidation products compared to the PTR-MS which we have used here. We have also used slightly different error types for PMF, which we showed to have a significant impact on the resolved factors, especially if the data size is small. Our results from PMF and NMF agreed reasonably well, even though NMF does not use an error matrix as an input, and it solves the bilinear equation with a different algorithm, indicating our PMF factorization is reasonable and correctly interpreted.

From a mathematical point of view, the selection of the most useful SDRT does not depend on the instrument used to measure the data, nor the extent of fragmentation taking place in the instrument. Only PAM is an exception here, as clustering techniques in general do not assign variables to multiple clusters (i.e., "between" clusters), whereas all the other presented SDRTs have the ability to share an ion between multiple factors. Similarly, if a large number of isomers is to be expected, NMF or PMF

may be preferable over EFA or PCA, as the latter two try to maximize the contribution of an ion to a single factor. Ultimately, however, the most useful choice of SDRT also depends on what kind of chemical processes are expected and measured, as the splitting of ions to multiple factors generally makes the interpretation of the factors more difficult, especially if the prevalence of possible isomerization is not known. Splitting of ions to multiple factors is also an important topic to discuss in source apportionment analysis, where an ion with specific m/z may emerge from various sources or source processes. However, it is a very subtle choice between possibly dismissing unexpected feature discovered by SDRT and using prior knowledge to validate the factorization results. Therefore, applying more than one SDRT may protect the user for determining surprising results "unphysical", and thus erroneous, but it also gives more robust outcome for the research when the results from different techniques agree.

## 6  Author's contribution

SI and SM designed the comparison study; EK and AV designed and organized the measurements/provided data; SI, LH, SM, and AB participated in data analysis and/or interpretation; SI wrote the manuscript; SM, AB, SS, LH, and AV edited the manuscript.

## 7  Acknowledgments

This work was supported by The Academy of Finland Centre of Excellence (grant no. 307331), The Academy of Finland Competitive funding to strengthen university research profiles (PROFI) for the University of Eastern Finland (grant no. 325022) and The Nessling foundation. Data collection for this study has been partly funded from the European Union's 10 Horizon 2020 research and innovation programme through the EUROCHAMP-2020 Infrastructure Activity (grant no. 730997).

Ville Leinonen is thanked for the help he provided with the R software and constructing the error matrices for PMF for the gas phase measurements.

## 8  Competing interest

The authors declare no competing interests.

## 9  Data availability

The data can be found in the EUROCHAMP database https://data.eurochamp.org/

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

**Tables**

**Table 1** Summary of the results for gas phase composition data. Best solution refers to the number of factors/clusters/components. The m/z refers to the mass with the $H^+$.

| Type of analysis | EFA | EVD-PCA | PAM | NMF | PMF (static error) | PMF (signal following error) | Example compounds |
|---|---|---|---|---|---|---|---|
| best solution | 4 | 4 | 4 | 5 | 5 | 5 | |
| rotation if used | Oblimin | Oblimin | - | - | - | - | |
| precursor (α-pinene) | 3 | 3 | 3 | 3 | 3 | 3 | α-pinene, $C_7H_{10}$, Toluene |
| early products | 2 | 2 | 2 | 2 | 2 | 2 | MVK, furan, acetaldehyde |
| later products/slowly forming products | 1 | 1 | 1 | 1 | 1 | 1 | $C_2H_2O$, $C_2H_8O$, $CH_3O_2$, MEK |
| intermediate products | - | - | - | 5 | 5 | 5 | Nopinone, m/z 157.08 |
| precursor (car exhaust) or background | 4 | 4 | 4 | 4 | 4 | 4 | $C_4H_8O_2$, m/z 167.06, Dimethylbenzene |

**Table 2 Summary of the results for particle phase composition data. Best solution refers to the number of factors/clusters.**

| Type of analysis | NMF | PMF (static error) | PMF (standard AMS error) | Example compounds |
|---|---|---|---|---|
| best solution | 4 | 2 | 2 | |
| primary OA (HOA) | 1 | - | - | m/z 12, 57, 59 |
| mixed LVOOA | 3 | - | - | m/z 44 |
| αP-SOA-SVOOA | 2 | - | - | m/z 43 |
| αP-SOA-LVOOA | 4 | - | - | m/z 44 |
| SVOOA | | 1 | 1 | m/z 43 |
| LVOOA | | 2 | 2 | m/z 44 |

5    **Table 3 The computational time (in seconds) for 2-9 factors for different SDTRs and data types and sizes.**

| SDRT | AMS | PTR-MS | PTR-MS*5 |
|---|---|---|---|
| EVD-PCA | - | 1.93 | 5.23 |
| SVD-PCA | 0.571 | 0.838 | 1.59 |
| ml-EFA | - | 14.2 | 16.9 |
| pa-EFA | - | 2.96 | 6.18 |
| PAM | 0.672 | 0.771 | 1.69 |
| NMF | 21.6 | 39.7 | 134 |
| PMF (static error) | 30.0 | 101 | 476 |
| PMF (standard AMS error or noise error) | 31.1 | 122 | 543 |

**Figures**

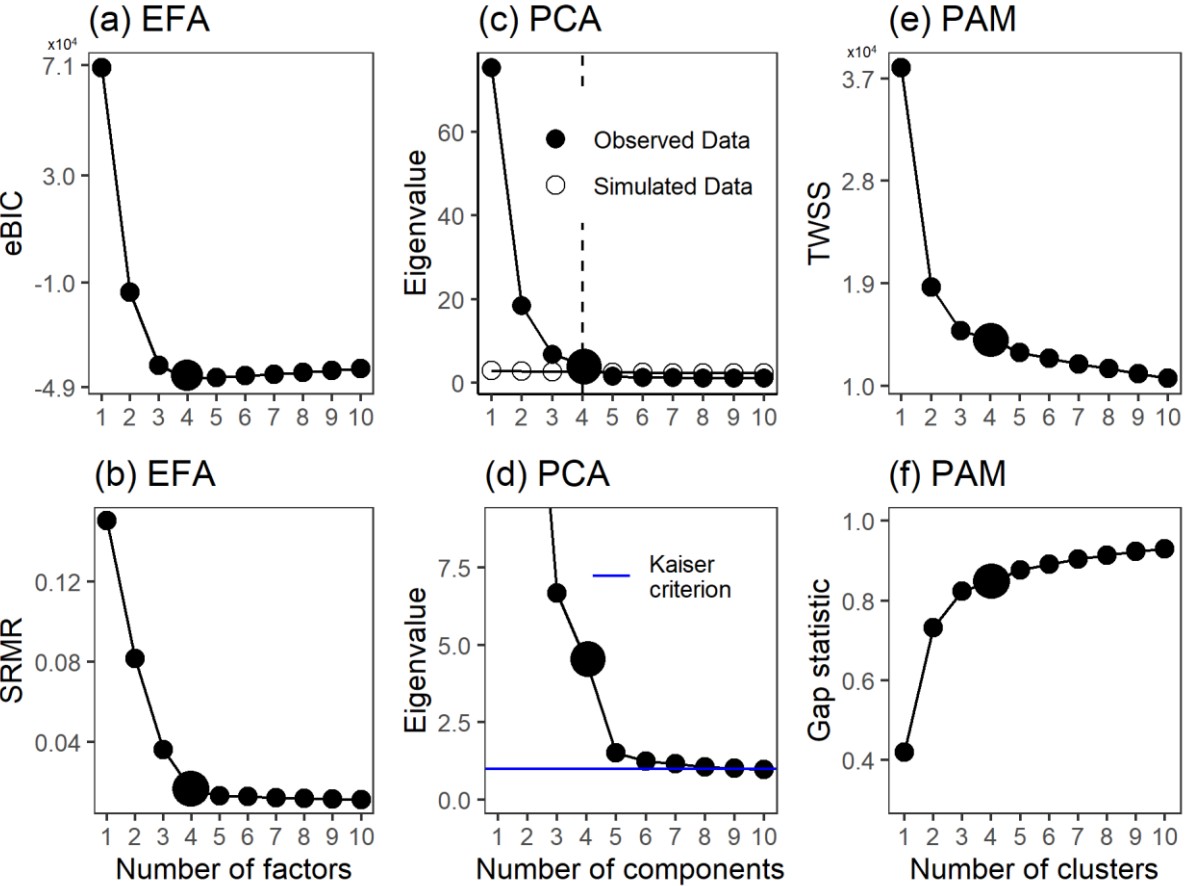

5      **Figure 1: Factor number indexes for gas phase data (PTR-MS). Empirical BIC (a) and SRMR (b) as a function of the number of factors for ml-EFA, Parallel analysis (c) and Kaiser criterion (d) for EVD-PCA and TWSS (e) and Gap statistic (f) for PAM. Larger points indicate the solution that was selected for more detailed interpretation.**

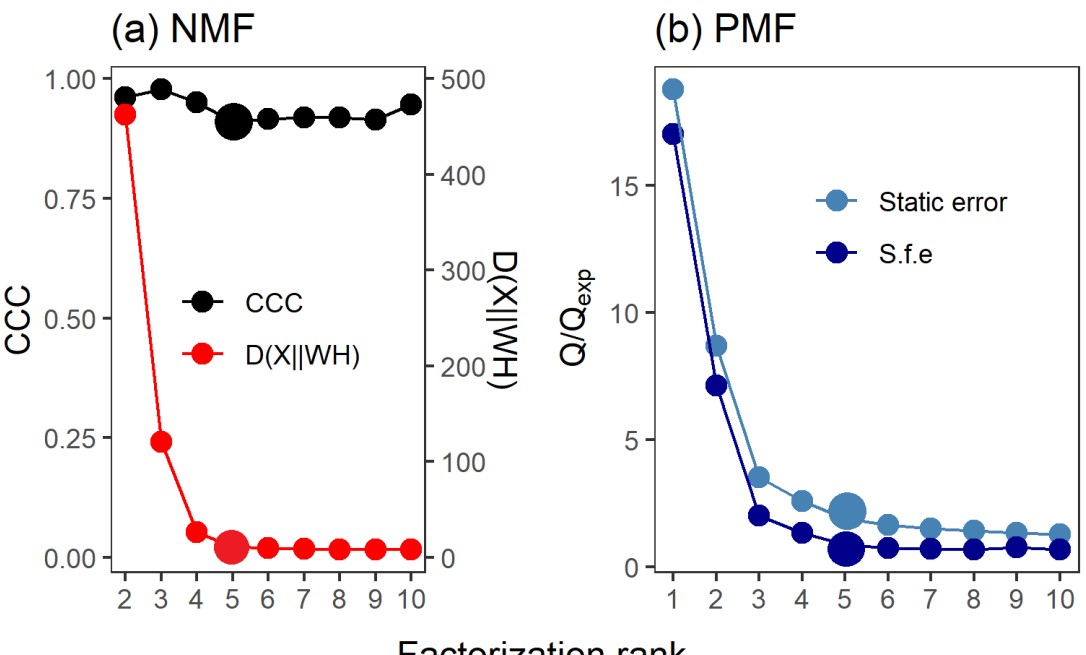

**Figure 2: Factor number indexes for gas phase data (PTR-MS). Estimation of the factorization rank for NMF in (a) with CCC and D(X||WH) and for PMF in (b) with $Q/Q_{exp}$ for the two error schemes (static error and signal following error, S.f.e). Larger points indicate the solution that was selected for more detailed interpretation.**

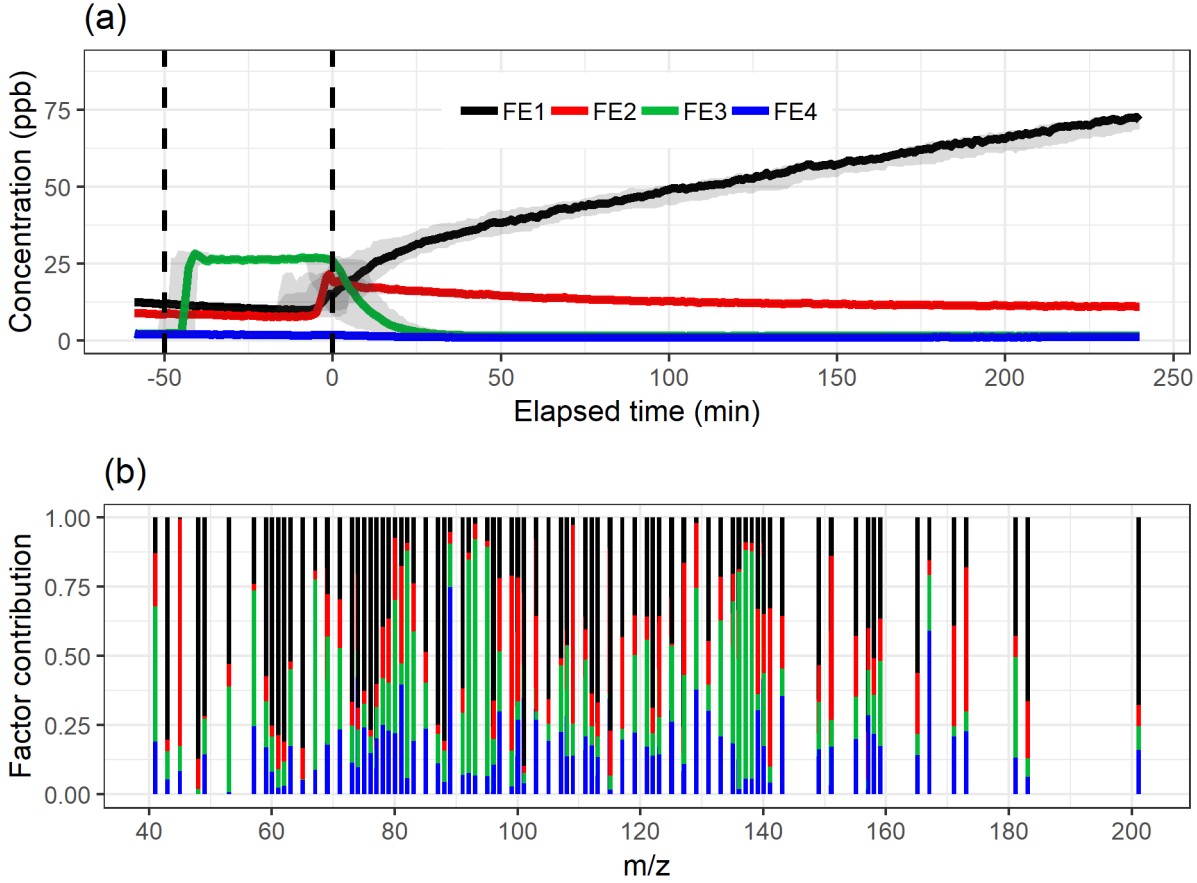

**Figure 3: The factor time series (a) and contribution (b) for ml-EFA with Oblimin rotation for the 4-factor solution. Shaded areas in the time series indicate the factor range for the bootstrap resamples, solid lines are for the measured gas phase (PTR-MS) data. The colour code identifying factors is the same in both panels. Factors were identified as later/slowly forming products (FE1), early products (FE2), α-pinene precursor (FE3) and background/car exhaust precursor (FE4).**

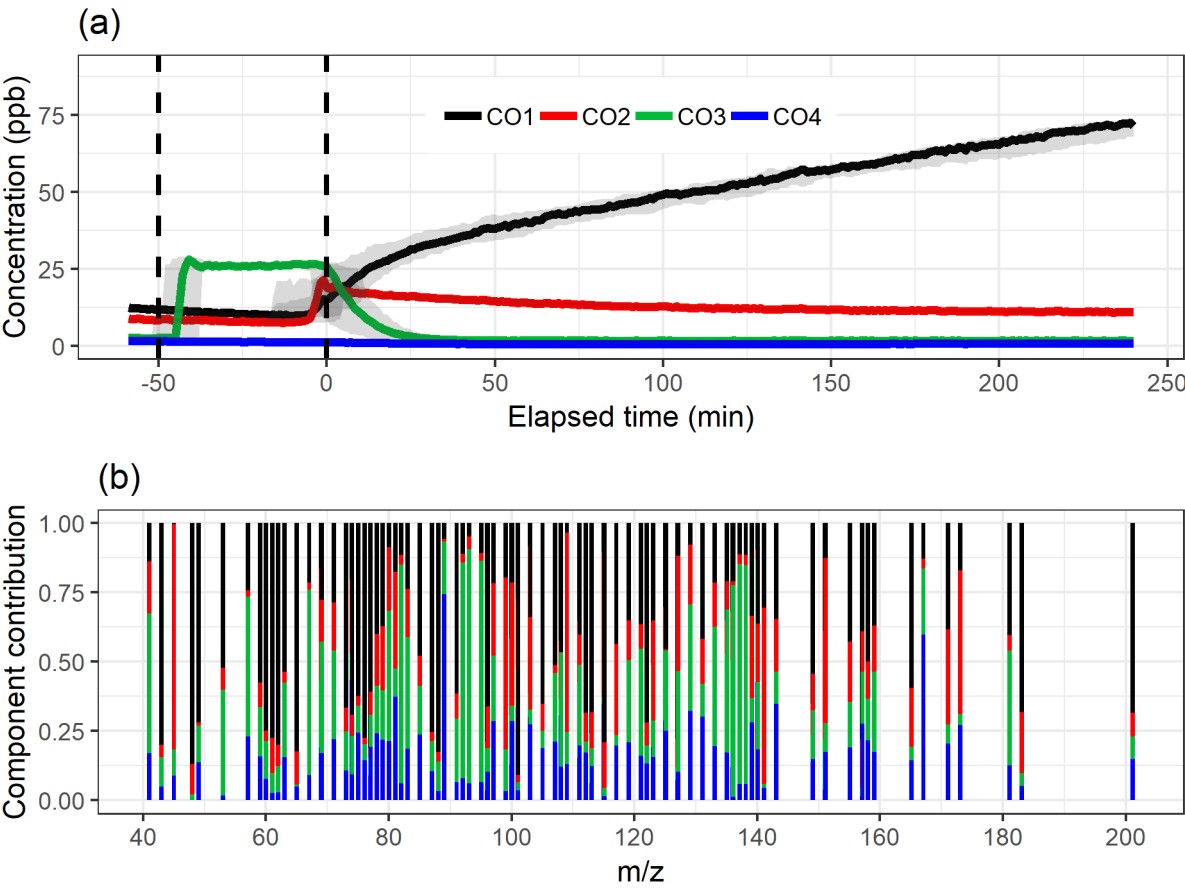

**Figure 4: The component time series (a) and contribution (b) for EVD-PCA with Oblimin rotation for the 4-component solution. Shaded areas in the time series indicate the component range for the bootstrap resamples, solid lines are for the measured gas phase (PTR-MS) data. The colour code identifying components is the same in both panels. Components were identified as later/slowly forming products (CO1), early products (CO2), α-pinene precursor (CO3) and background/car exhaust precursor (CO4).**

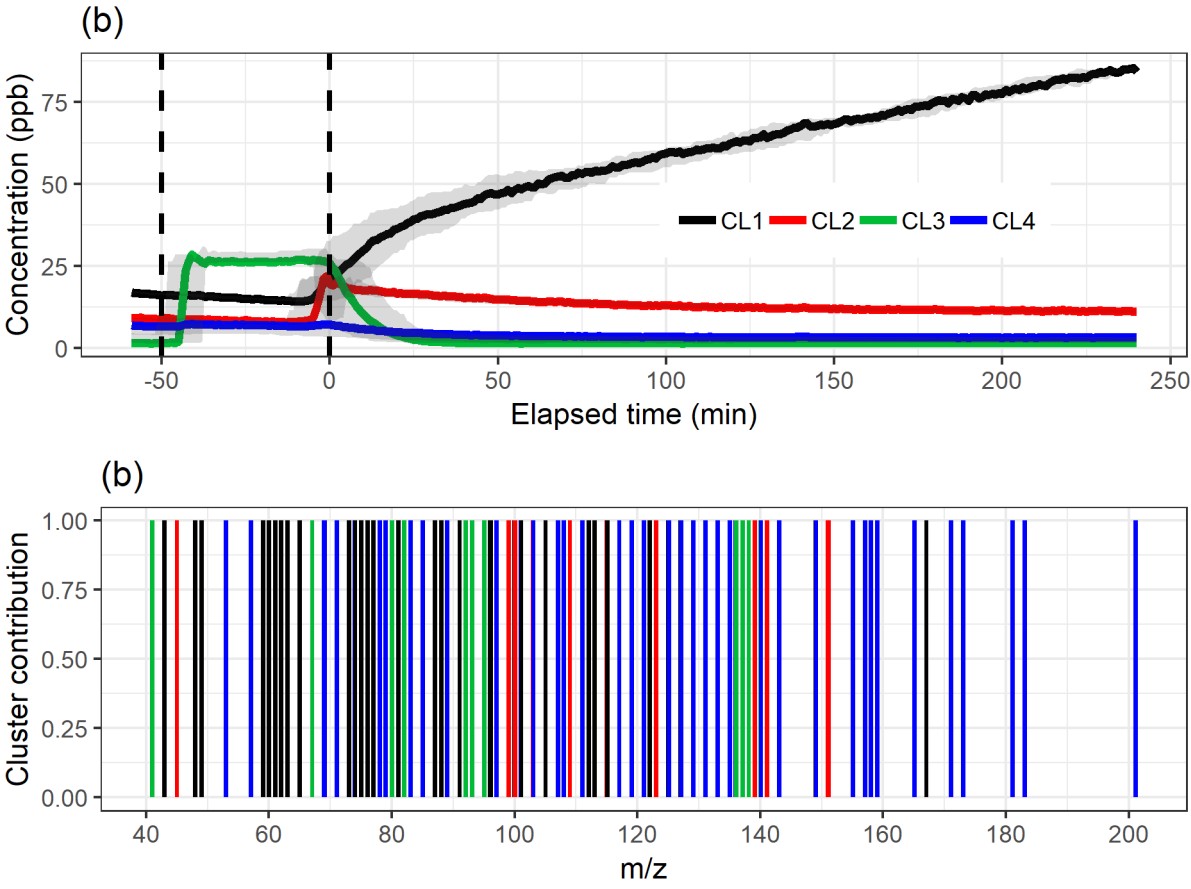

**Figure 5: The time series (a) of the clusters and the distribution of ion to clusters (b) from PAM with 4-cluster solution. Shaded areas in the time series indicate the cluster range for the bootstrap resamples, solid lines are for the measured gas phase (PTR-MS) data. The colour code identifying clusters is the same in both panels. Clusters were identified as later/slowly forming products (CL1), early products (CL2), α-pinene precursor (CL3) and background/car exhaust precursor (CL4).**

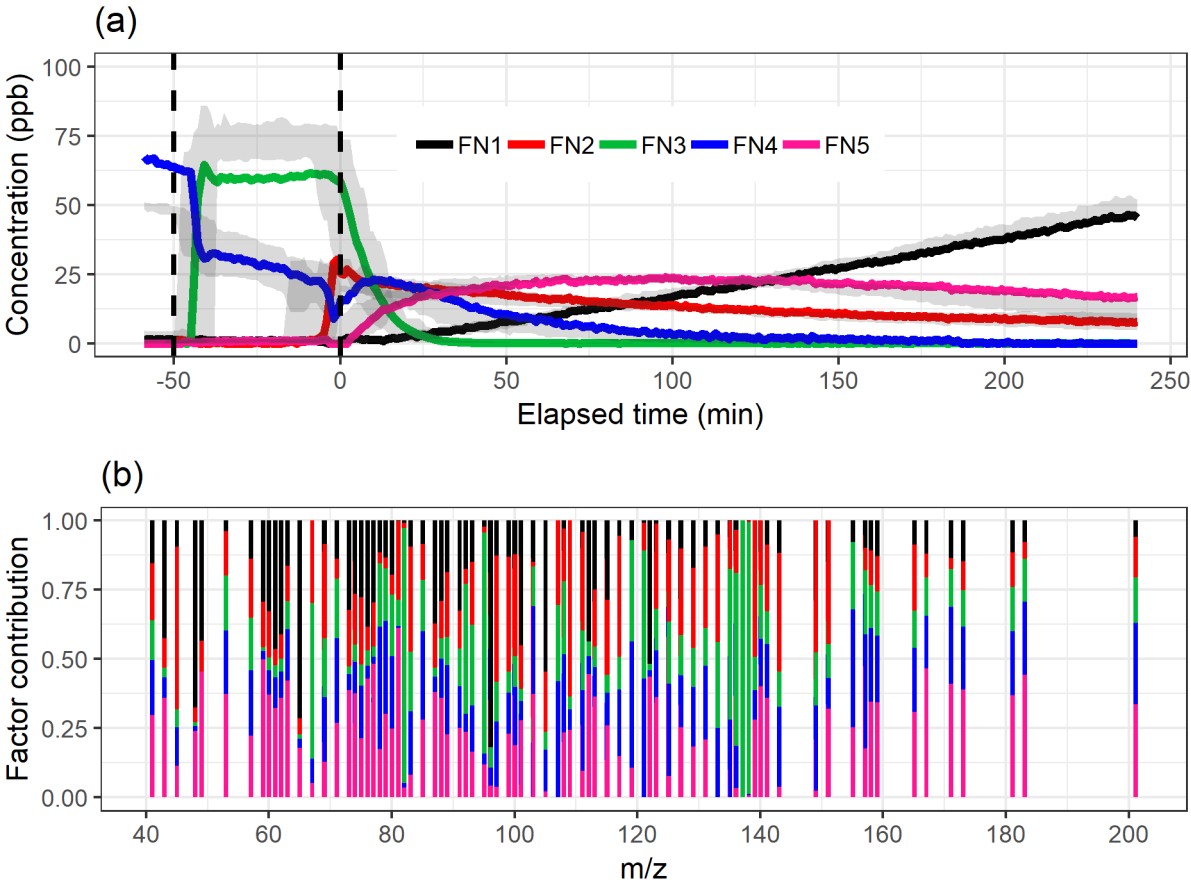

**Figure 6: The time series (a) of the factors and the distribution of ion to factor (b) from NMF with 5-factor solution. Shaded areas in the time series indicates the factor range for the bootstrap resamples, solid lines are for the measured gas phase (PTR-MS) data. The colour code identifying clusters is the same in both panels. Factors were identified as later/slowly forming products (FN1), early products (FN2), α-pinene precursor (FN3), background/car exhaust precursor (FN4) and intermediate products (FN5).**

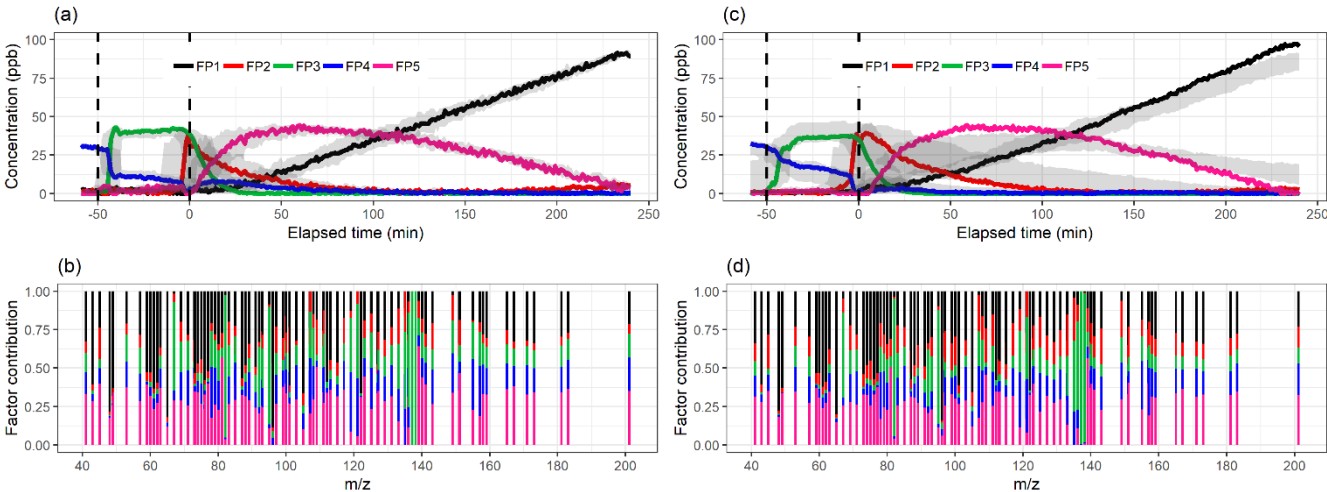

**Figure 7: Factor time series and contribution from PMF with static error (a-b) and signal following error (c-d) for factorization rank 5. Shaded areas in the time series indicates the factor range for the bootstrap resamples, solid lines are for the measured gas phase (PTR-MS) data. The colour code identifying the factors is the same in the top and bottom panels. Factors were identified as later/slowly forming products (FP1), early products (FP2), α-pinene precursor (FP3), background/car exhaust precursor (FP4) and intermediate products (FP5).**

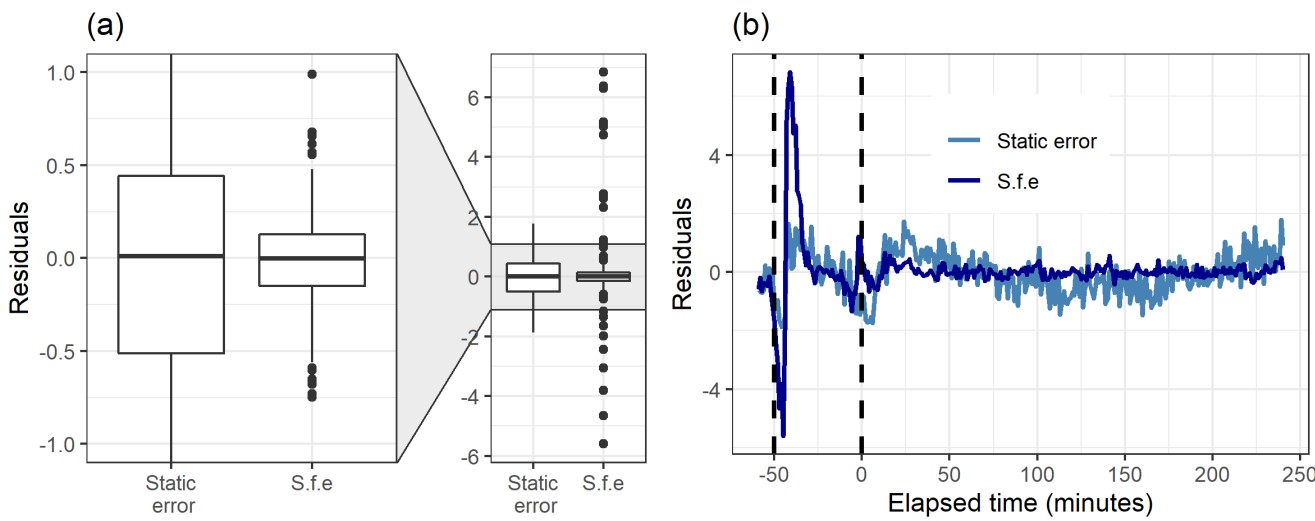

**Figure 8: Boxplot (a) and the time series (b) of the residuals (original total signal – reconstructed total signal) with static error and signal following error (S.f.e) with 5 factors from PMF for the measured gas phase (PTR-MS) data.**

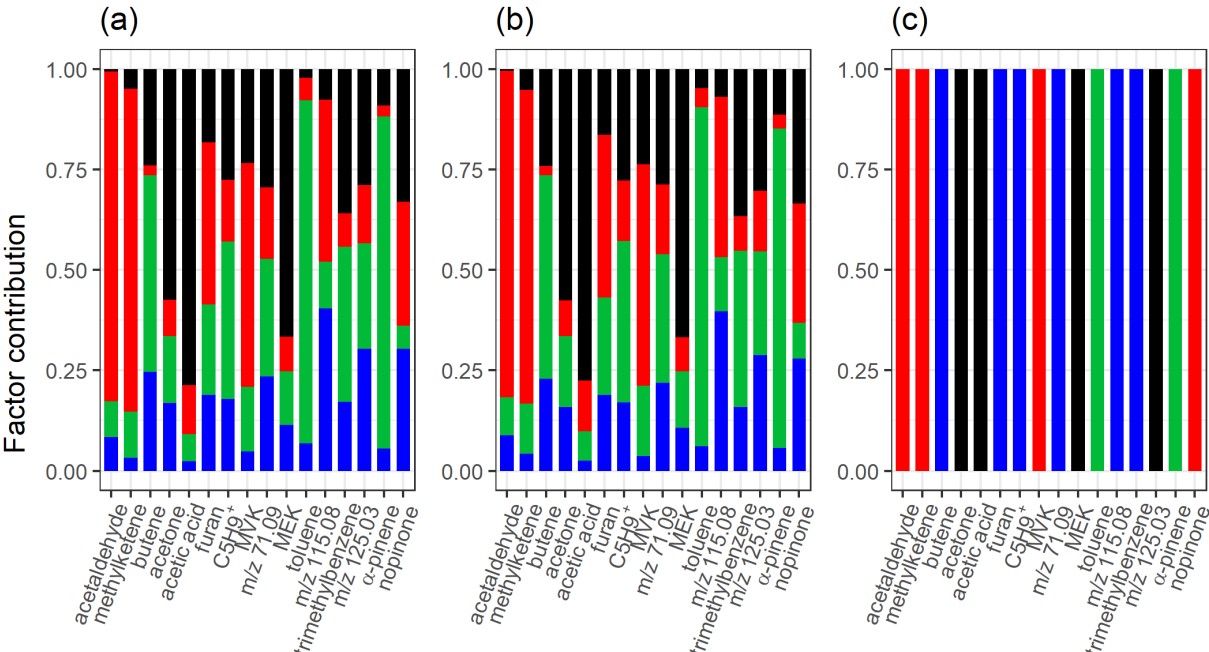

**Figure 9: Total factor/component/cluster contribution of selected compounds from a) ml-EFA, b) EVD-PCA and c) PAM for the measured gas phase (PTR-MS) data. Colour code identifying the factors/components/clusters is the same in all panels and were identified as later/slowly forming products (black), early products red), α-pinene precursor (green) and background/car exhaust precursor (blue).**

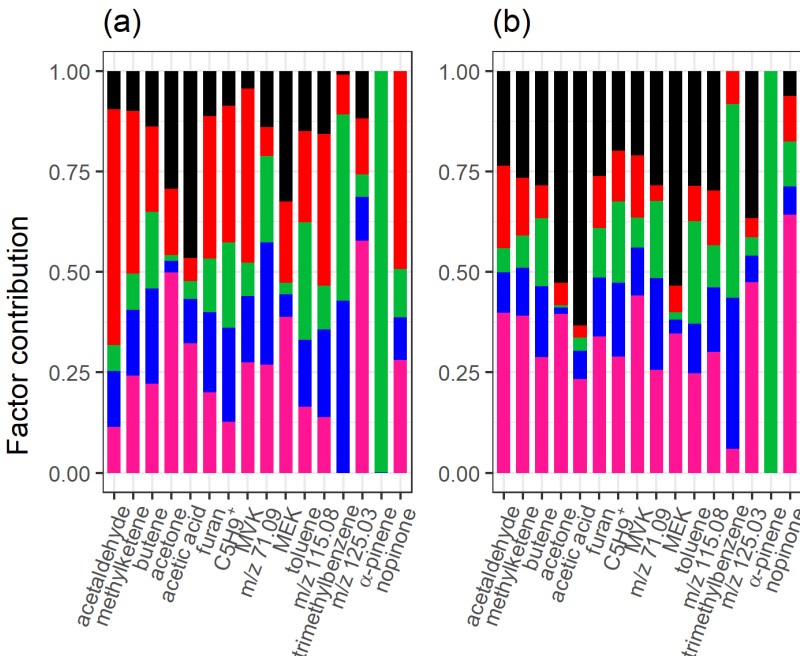

**Figure 10: Total factor contribution of selected compounds from a) NMF and b) PMF with static error for the measured gas phase (PTR-MS) data. Colour code identifying the factors is the same in both panels and were identified as later/slowly forming products (black), early products red), α-pinene precursor (green), background/car exhaust precursor (blue) and intermediate products (pink).**

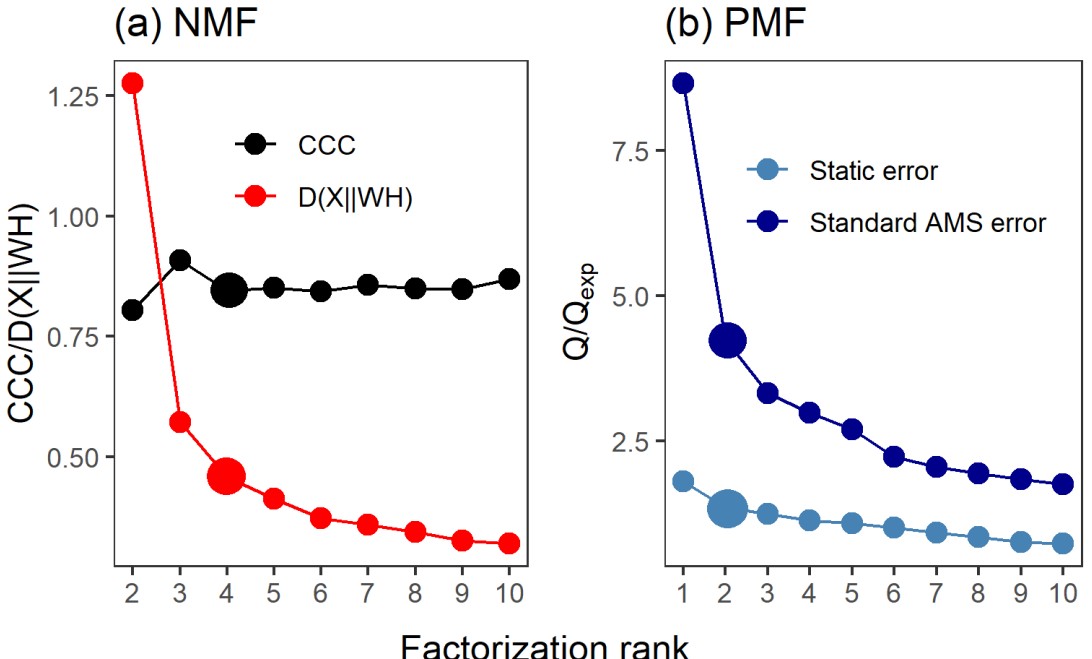

5    **Figure 11: Factor number indexes for particle phase data (AMS). Estimation of the factorization rank for NMF in (a) with CCC and D(X||WH) and for PMF in (b) with $Q/Q_{exp}$ for the two error schemes. Larger points indicate the solution that was selected for more detailed interpretation.**

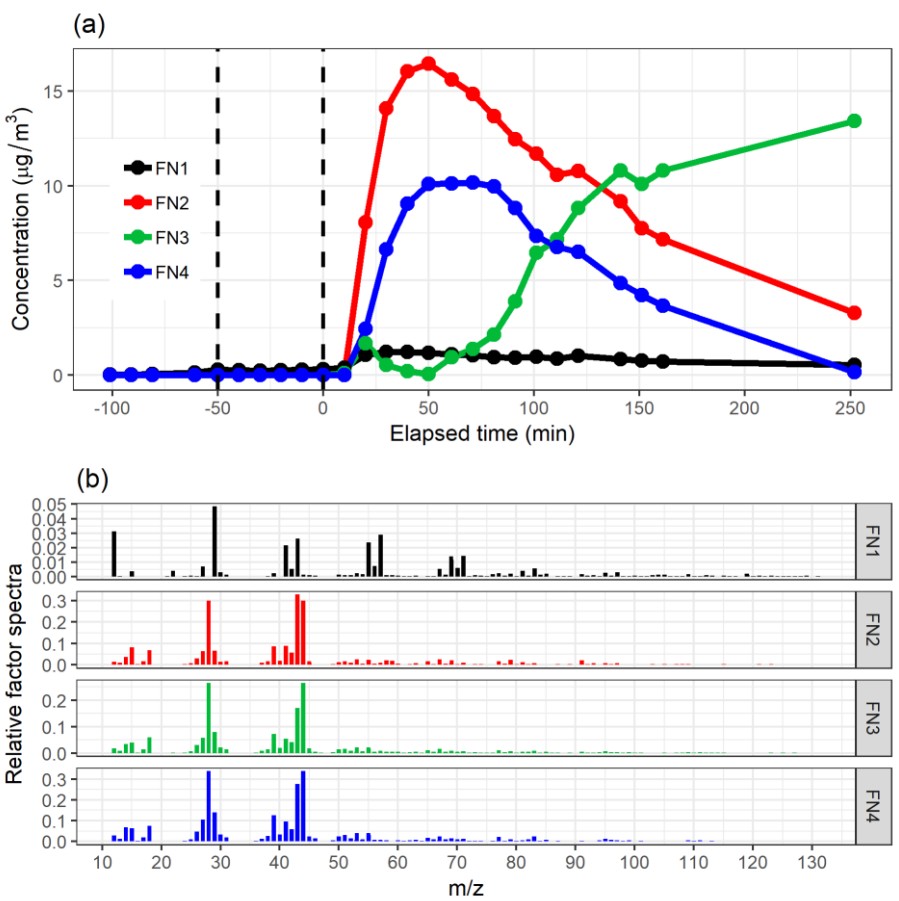

**Figure 12: The factor time series (a) and relative factor spectra (b) from NMF with 4 factors for the measured particle phase (AMS) data. The colour code identifying the factors is the same in both panels. Factors were identified as primary OA (FN1), αP-SOA-SVOOA (FN2), mixed LVOOA (FN3) and αP-SOA-LVOOA (FN4).**

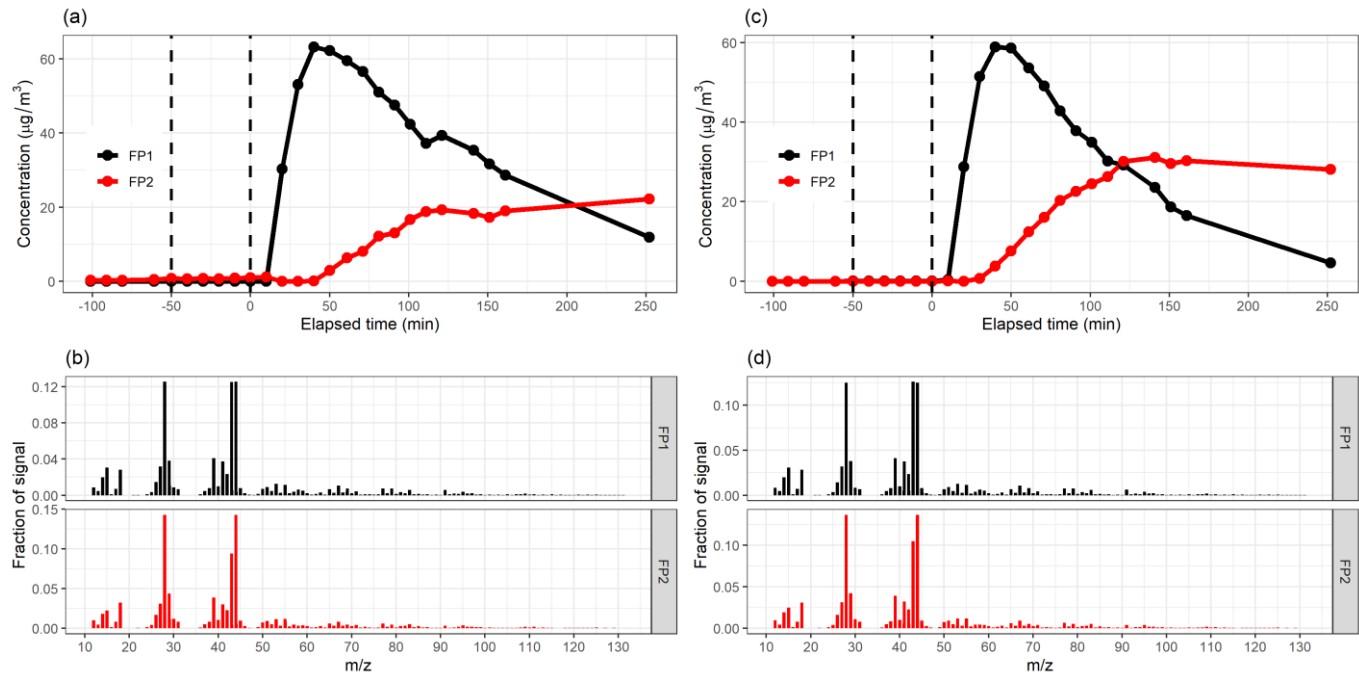

**Figure 13: Factor time series and contribution from PMF with static error (a-b) and standard AMS error (c-d) for factorization rank 2 for the measured particle phase (AMS) data. Factors were identified as SVOOA (FP1) and LVOOA (FP2)**

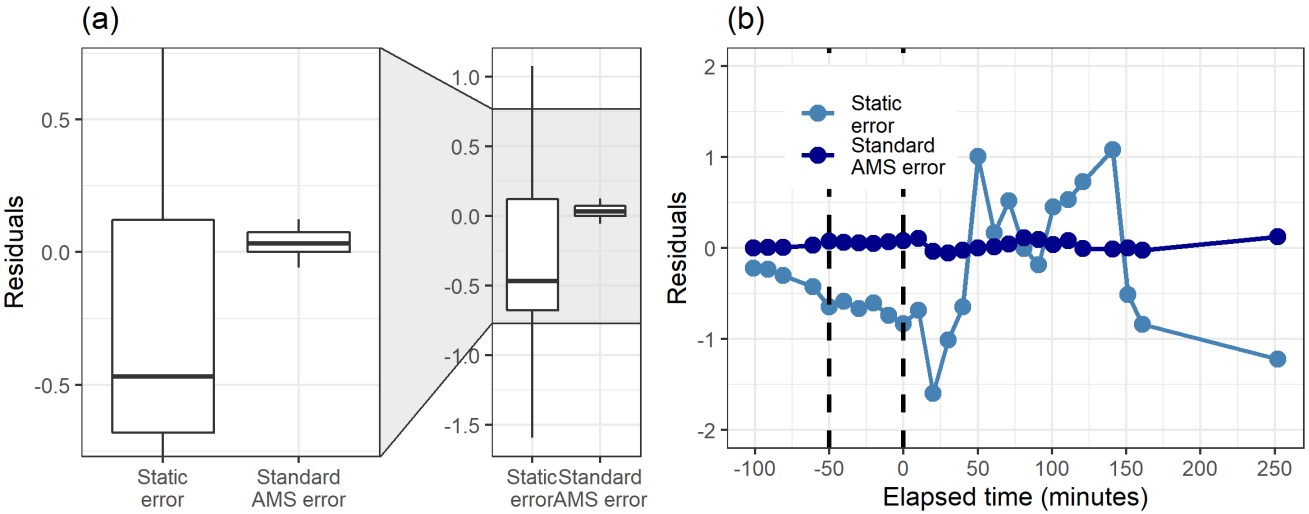

**Figure 14: Boxplot (a) and the time series (b) of the residuals (original total signal – reconstructed total signal) with static error and standard AMS error with 2 factors from PMF for the measured particle phase (AMS) data.**

**Appendix A**

**Table A1 Mathematical symbols and notations used in the equations throughout the paper.**

| symbol | explanation |
|---|---|
| $\mathbf{X}$, $X_{ij}$ | data matrix (n x m), data matrix element |
| $p$ | number of factors/components/clusters |
| $\boldsymbol{y_j}$ | variable/ion $j$ (time series vector), column $j$ from $\mathbf{X}$ |
| $\boldsymbol{c_j}$ | PCA component $j$ |
| $\boldsymbol{f}$ | EFA factor |
| $\boldsymbol{\lambda}$, $\lambda_{ij}$ | EFA loading matrix, loading matrix element |
| $\mathbf{S}$, $\mathbf{R}$ | observed correlation matrix, implied correlation matrix |
| $\mathbf{G}$ | factorization matrix (factor time series) PMF (n x p) |
| $\mathbf{F}$ | factorization matrix (factor spectra/contribution) in PMF (p x m) |
| $\boldsymbol{\mu}$ | PMF error matrix |
| $\mathbf{E}$ | residual matrix in PMF |
| $\mathbf{W}$ | factorization matrix (factor time series) in NMF (n x k) |
| $\mathbf{H}$ | factorization matrix (factor spectra/contribution) in NMF (k x m) |