# Peer review of "Comparison of dimension reduction techniques in the analysis of mass spectrometry data"

_Atmospheric Measurement Techniques, 2019_

## Referee Comment (RC1)

**Review of Comparison of dimension reduction techniques in the analysis of mass spectrometry data**
**Atmospheric measurement techniques, AMT-2019-404**
**Isokääntä et al.**

**Summary:**

This work accomplishes a cross-comparison of several data-reduction analysis techniques. The analysis is performed on a chamber experiment. Car exhaust was directly sampled into an environmental chamber. A-pinene was also added to the chamber. The mixed car exhaust/pinene was then aged via OH-initiated photooxidation. Two instruments were used, a PTR-ToF MS and an AMS. Data were then analyzed using principal component analysis, positive matrix factorization, exploratory factor analysis, clustering, and non-negative matrix factorization. The resulting simplifications are compared in terms of the ability to reconstruct the original data set (residual), number of factors/groups required to explain data variability, time-series behavior of the factors, and chemical composition of the factors. The authors find that the preferred number of factors is roughly similar regardless of technique. Some factors (or groupings) are generally consistent (in terms of time-series behavior and composition) regardless of technique. Some techniques, particularly PCA and EFA were found to be difficult to interpret and not as useful. The clustering method was not useful for AMS data.

**Major comments:**
The manuscript is generally well-written and organized. The sampling techniques and experimental conditions are appropriate.

This work presents a clear, helpful, and timely cross-comparison of results between different analysis methods. Because modern analytical instrumentation produces large datasets, computer assisted analysis is unavoidable. A major question in the field is if the results are robust, reproducible, etc. That the same major patterns and groupings appear (with minor differences) regardless of which technique is used is an important result that provides legitimacy to the vast amount of work currently being done with PMF and other techniques in atmospheric chemistry. Therefore I recommend this paper for publication.

My major questions include the following.
1) The pre-light mixing period is a substantial fraction of the whole experiment- is this interesting? The inclusion of this time period seems to have significant impact on the algorithm results.
2) Mostly this paper seems to show that various different algorithms distinguish similarly between primary and secondary VOCs. Is this a useful reduction of chamber data, to group all secondary VOCs into one or two blocs? From Table 1 it seems that the two matrix factorization techniques result in perhaps three oxidized factors, but this is hardly discussed in the text. It is not clear to me if the three oxidized factors are consistent between the two techniques. It would be helpful to have more interpretation of these factors, especially if it were supported by a more detailed connection to the chemistry of the system.
NMF also seems to result in some mixing of primary and secondary emissions (FN4) which is probably unphysical.
3) The way the mass spectra (chemical composition of groupings) are presented is very difficult to interpret and compare. From Table 1 the authors have assigned a consistent identity to factors resulting from each technique. Can you show a direct comparison of "Factor 2" for example, perhaps by plotting one mass spectra against the other so that it is easy to see which VOCs are similarly enhanced, and which may be different?

**Specific comments:**

Page 3, line 3-4: I disagree with this statement, "PMF was originally developed for field measurement data sets where real changes in factors are expected to be much slower than e.g. the noise in the data." In the ambient environment, VOC composition and concentrations can actually change very quickly (on the order of seconds), especially when plumes of highly-concentrated primary emissions are intercepted. The abrupt changes in conditions during a chamber experiment therefore do not present a special challenge to PMF, compared to ambient measurements.

Page 3, lines 26-29: A few more details are needed here (instead of in the supplement):
What was the typical concentration of total VOC (or of a few key VOC e.g. aromatics)?
What concentration of α-pinene was added?
What was the VOC-NOx ratio, and how was it adjusted?
Why were these specific concentrations of vehicle exhaust VOC and α-pinene chosen?
Were vehicle exhaust and pinene added just at the beginning of the experiment, or were they continuously injected?
Was the chamber continuously refilled (and with clean air or with fresh emissions?) to replace air taken by the mass spectrometers, or did the volume of the chamber decrease over time?

Section 3.1.2: EFA seems very similar to PMF. Could you please explain the major relevant difference(s) between EFA and PMF, and how they would affect the resulting dimensionality reduction?

Page 7 line 16 (and elsewhere): At multiple points in the manuscript it is mentioned that the rapid changes associated with lights-on cause problems when implementing the various dimensionality reduction techniques. Would it make more sense to exclude data prior to t=0? Was there a reason this was not done?

Page 10 lines 28-31: Since Figure 1 relies on a comparison of BIC and SRMR, it would be helpful here to provide more detail on how these two metrics are calculated, what the relevant differences are, and why one may be preferred over the other. What was the purpose of calculating both metrics and why were these particular metrics chosen?

Equations 11 and 12: There are several errors in these equations which are likely a copying error from Brunet et al. 2004. The authors should check that the actual implementation was done correctly. The equations should read:
$$H_{au} \leftarrow H_{au} \frac{\sum_i W_{ia} X_{iu}/(WH)_{iu}}{\sum_k W_{ka}}$$
$$W_{ia} \leftarrow W_{ia} \frac{\sum_u H_{au} X_{iu}/(WH)_{iu}}{\sum_v H_{av}}$$

I also suggest here to use "k" as the row index for **W** (in the denominator term), to avoid confusion with "p" being the factor rank, and for consistency with Lee and Seung, 2001.

Page 11 lines 29-33: Given that the update functions (11) and (12) are derived from the divergence cost function $D(X||WH)$ (Lee and Seung, 2001, Eq. 3), I suggest that this cost function is monitored as a function of $p$, analogously to $Q/Q_{exp}(p)$ for PMF. The termination condition for NMF wasn't described in Section 3.1.4, but presumably it is not dependent on $p$; if this is the case then the divergence of the end solution can be compared for each value of $p$.
The residual sum of squares is not an appropriate metric, as this was not the cost function used for the NMF implementation.

Page 12 line 2: What is meant by "not achieved only by change?"

Page 13 line 5: Why not compare the absolute value of the residual?

Pages 14 and 15: For other researchers which would like to use this paper as a guide, it would be helpful to indicate the range of values that are acceptable. For example Page 14 line 4-5, what value of residual would be considered not acceptable? Page 14 line 32, what is the Kaiser limit and what range of values are considered "close"? Page 15 line 11 are these considered large or small residuals?

Page 14 line 1 and page 15 line 2: Can you show please how it is determined that the additional component is not a new component with different properties but rather a mixture of previous components?

Page 17 Lines 16-17, 23-30: The signal following error is essentially introducing a smoothness constraint, which doesn't seem appropriate given that you know there are sharp changes due to experimental conditions. Is it recalculated for each data resampling? Why not resample the error along with the data? Is it possible to split the data into time periods whose start and stop are defined by sample injection, lights on, etc. so that the running standard deviation does not include these sharp changes?
Additionally, lines 26-27: Ambient data often has fast changes that are due to real variability. This is one of the reasons why fast online techniques such as PTR-MS are used for ambient measurements, because they allow the observation of these changes.

Page 22 line 20: This indicates that the error estimation should be revised, or that these compounds should be downweighted. PMF and NMF are extremely similar techniques with the crucial difference being only the inclusion of an error matrix, so it does not seem likely that the difference in performance is due to the size of the dataset. The extremely small values of NMF residuals also seem suspect. The authors should check that residuals for PMF and NMF were calculated in exactly the same way so as to enable the direct quantitative comparison.

Page 24 line 13-14: Is this correct? I read this paper as well. I thought they had PTRMS and AMS. Additionally, PTR-TOF is a subset of TOF-CIMS, or?

Figures 3,4: In the plot caption or legend it would be helpful to have a brief description of the interpretation of each factor, e.g. "pinene", "car exhaust", "background". Additionally the display in plot (b) of factor contribution as a function of m/z doesn't add much to the paper; I wouldn't expect the factor contribution to depend on m/z in any particularly meaningful way. Since (I believe) your PTRMS has multiple peaks resolved at each nominal mass, showing a unit-mass stick spectrum here is also not especially meaningful. If you want to show mass spectra I strongly suggest to break panel b into 4 separate spectra, one for each factor, so that they can be examined separately.

Figure 9: Where does isoprene come from in this experiment? Is this more likely to be a hydrocarbon from vehicle exhaust? Cycloalkanes in fossil fuel are known to create PTR ions at $C_5H_9+$, see e.g. Yuan et al. Chem Rev. 2017 doi.org/10.1021/acs.chemrev.7b00325.

Figure 13: The factor time series for the most part do not look realistic. What physical process could lead to the non-smooth behavior and multiple maxima?

**Minor/Technical corrections:**

Page 2 line 22, "alike" -> "like"

Page 3 line 22, "was" -> "were"

Page 13 line 24, "described in section 0" -> "described in section 3"

Page 15 line 19, "gab" - > "gap"

Page 19 line 27, "much" -> "many"

---

## Referee Comment (RC2) · M. Äijälä (Referee) · 25 Feb 2020

Review of S. Isokääntä et al. (AMT-2019-404):

**Comparison of dimension reduction techniques in the analysis of mass spectrometry data**

**Summary**

Sini Isokääntä and co-authors present a comparison of dimensionality-reductive techniques for mass spectral data analysis, applied to a data set of car exhaust + a-pinene aging experiment in a reaction chamber. The data, collected by a PTR-MS and an AMS, was factorized and clustered using 5 different techniques. The authors present and discuss their analysis procedures and interpretation of results from mathematical and physicochemical viewpoints. For the PTR-MS data, all five techniques produce comparable results, yielding 4 to 5 interpretable factors (or clusters). This is a very important result considering the novelty of applying these methods to PTR-MS and similar mass spectral data of gas phase. For AMS data, only PMF and NMF yielded results that could be compared – PCA, EFA and PAM struggled with the particle data set. Although applied to data from a single experiment, the comparisons presented and the discussion on analysis techniques convey general messages for many similar analyses that abound in atmospheric science.

The manuscript is clearly of interest to AMT readers, and touches the important topic of computational analysis of complex, large mass spectrometric data sets and advanced statistical methods of data analysis. Despite the evident need, a very limited amount of reviews of advanced statistical methods for this type of complex, physicochemical analyses is available, so I see the authors' contribution as extremely welcome to the field. The manuscript is well and clearly written and structured. With some exceptions, detailed in specific comments, the data, analysis techniques and experimental setup is adequately described. I recommend the paper for publication in AMT after addressing the following comments.

**Major comments**

The authors generally do a good job of introducing and describing their methods, but I would call for discussion on the physicochemical *objectives* or *purpose* of the statistical techniques used in this study. Obviously it is about more than reducing the size or dimensions of the data, which is the single common feature of all these SDRTs. However, some of these methods are rather similar (e.g. PMF vs NNMF) while some aren't (e.g. PAM vs factorization methods). Generally, a data analyst should choose a proper tool for the job, depending on the goal. Here it seems the methods are just applied on the data, seemingly without stating beforehand e.g. the difference in categorizing or classifying variables using PAM, studying the main explanative components of variability or doing latent feature extraction by exploratory factorization. Many of these differences are later casually mentioned, but I suggest an effort to further summarize these fundamental differences, and conversely, some of the close similarities could be made in the introduction and conclusions of the paper.

Secondly: algorithmic methods are often discussed plenty in these types of reviews, but some of the equally significant, but less obvious issues include, among others: data pre-processing (e.g. scaling and weighting), metrics (e.g. for quality evaluation and similarity), and error models. Mostly, the authors cover these topics well enough, but some major questions surrounding e.g. the non-standard PMF error model the authors used and preprocessing of data for PCA/EFA are raised. See specific comments on the details.

Finally, most factorization methods, including those used in this paper, assume constant factor profiles, i.e. that chemical reactions do not take place. This is fundamentally at odds with the approach of using them to model data of reaction chamber chemistry. In practice, the methods *can work* despite this (as is shown in this work), but this fundamental violation of basic assumptions should be explicitly stated and discussed.

**Specific comments**

**p.2.l.2.:** "…sharp change at the beginning of experiments, e.g., switching on UV-lights) may present additional challenges, as PMF was originally developed for field measurement data sets where real changes in factors are expected to be much slower than e.g. the noise in the data.". In my understanding PMF is not assuming anything about order of measurement points or rates of change in loadings – please provide a reference or expanded rationale for this line of thinking!

**p.2.l.14.** "In our study we chose a set of SDRTs having fundamental differences." Aside from data reduction or simplification, the objective of SDRTs differ. Often, clustering is applied to classify or categorize non-correlated variables, whereas factor analysis aims to uncover latent features the combination of which would best explain the variation in data. Can the authors please comment on their *objective of the analysis* outside of reduction of data size) of the use of SDRTs for this type of analysis? Especially regarding clustering.

**p.2.l.16:** "On the other hand, clustering might be more suitable for a more simplified or preliminary approach, or when the chemical compounds in the data are already known." It is not trivial to envision a case where clustering would be preferable, even as a preliminary approach – please expand on this reasoning or provide an example.

**p.3. :** Please describe the PTR-MS and the SP-AMS in some more detail here. Specifically their mass analyzer resolution (C/HR/L –ToF), as this strongly affects mass analysis and type of data.

**p.3.l.9.**: A note. As the authors state, if one does not account for mass spectra baseline, this may give rise to "background factors" or cause variables to get polluted or mixed (depending on instrument resolution). While theoretically these could be separated by factor analysis, in practice their presence in data hinders the analysis, often notably. For hard clustering, this effect of may even more problematic. Can you give an order of magnitude estimate of this effect? Can you separate baseline factors / clusters in some cases? What is their fraction of total explained variability?

**p.3.l.18.:** This type of truncation correction is likely detrimental to analysis and should be avoided in statistical analysis (especially factor analysis, as it creates an artificial, non-random, variability component). If only few points had this problem, would it not be preferable to omit them?

**p.5.l.30:** "[…]*where X includes the analysed variables (centred and scaled by their standard deviations)*" In their examination of errors in PCA (and PMF) in environmental applications Paatero and Hopke (2003) strongly advice against "autoscaling":

> *"5.1. Do not autoscale noisy variables in PCA*
>
> In PCA, it is customary to scale columns of **X** so that in the scaled matrix, all of the columns have the same variance. This procedure means that the sum of the two components of the variance (signal and noise) is constant over all variables. It follows that for the weakest variables, having the smallest amount of signal, the noise variance is much larger than for the strong variables. This behavior is in severe conflict with the recommendations found in this work: the exaggerated noise in a few noisy variables will cause the small principal components to be undetectable in the analysis and will increase the noise in other principal components. The recommendation is clear: "do not auto-scale noisy variables in PCA modeling" […]
> .

Is this type of scaling also done in here? Please reflect and add some discussion on the issue of error model in PCA. Perhaps recommend the readers to note this for future analyses?

**p.5.l.24. :** While potentially simplifying the chemical interpretation of factors, rotating the variables in a direction where ions are more separated between the factors, does this not equally degrade the time series interpretability (loadings' time series get more similar)?

**p.6.l.14.:** Same question on EFA error model as for PCA above, as PCA and EFA are very similar methods. Please discuss error weighting and accounting signal-to-noise (Paatero and Hopke, 2003).

**p.6.l.6.:** Does the limit of 0.3 apply to AMS data as well? This means any m/z signal explaining under 0.3 ug/m3 was set to zero? How many variables does this affect – I would imagine it is a very large fraction? How does this truncation affect mass (signal) conservation in data?

**p.7.l.15** On the error model of st.dev / sqrt(n): as written in the text, st.dev reflects the changes in the concentrations of compounds in the chamber, during the experiment, and not measurement uncertainty (or counting statistics error of the instrument) in a repetition measurement? How does dividing it by (square root of) data series length make it a more relevant error metric? Please explain. Also: Why not use the standard error model computed by the AMS analysis software (Squirrel/Pika; e.g. Ulbrich et al., 2009)?

**p.7.l.21.** (and in the supplementary referred): In calculation of the signal-to-noise-ratio (SNR), you identified weak and bad variables (defined by their low SNR). However in contrast to usual practice (in PMF), you do not down-weight these signals. Especially since you are using a custom (not thoroughly validated) error model, I would listen by the advice, Paatero and Hopke (2003):

> *"Regarding weak/bad variables, the main result of this work is that even a small amount of overweighting is quite harmful and should be avoided. In contrast, moderate downweighting, by a factor of 2 or 3, never hurts much and sometimes is useful. Thus, it is recommended to routinely downweight all weak variables by a factor of 2 or 3. This practice will act as insurance and protect against occasions when the error level of some variables has been underestimated resulting in a risk of overweighting such variables. Regarding bad variables (where hardly any signal is visible from the noise), the recommendation is that such variables be entirely omitted from the model."*

The reasoning for not down-weighting only states that low SNR data has high noise-to-signal rate, which is only stating the evident, and not very useful. Also, the small number of variables is not really a good excuse to deviate from the practice (without at least some sensitivity analysis). While I agree the overall error from not doing this properly is likely smallish (for PMF), I recommend you acknowledge the issue and cite this "best practice", the Paatero and Hopke (2003) recommendation, for the readers – not to proliferate a deficient data pre-processing practice.

**p.7.l.26.,** also Figure S4 (should be: "S7"?). The standard AMS error model is composed of a minimum (Gaussian) error (Ulbrich et al., 2009), related to electrical background noise of the instrument (background at zero signal) plus a counting statistics uncertainty that follows the Poisson distribution (error is proportional to the square root of signal intensity; Allan et al., 2003). This model seems different from the ones used in this paper. Please comment on this, and why you chose not to directly use the approach of Allan et al and Ulbrich et al. (2009), readily available from the AMS analysis software.

From the plot S4 (should be "S7"?) "PMF error schemes" it seems the constant error scheme ("Static error") overestimates error, especially for low concentrations, and the counting statistics (Poisson) error seems negligible (underestimated), even for the most abundant aerosol ion, at m/z 44, at highest concentrations. Please double check and report the error calculations here against what you get from the "Squirrel" AMS analysis software.

Why was the "signal following error" not used/tested for the AMS, since it seems to work for PTR-MS?

**p.8. l.6.:** How much does the Q (or Q/Q exp ratio) increase with these fpeak values?

**p.9.l.10.:** How do you interpret the negative loadings in EFA and PCA? Why were these solutions deemed physically sound, if they feature negative mass loadings?

**p.10.l.5.:** *"Depending on the aim of the study 5 and the type of the data, this property of cluster analysis may be considered either as an advantage or disadvantage."*. This is an important point in this paper overall, as is the difference between hard vs soft divisions of variables. Please elaborate, and e.g. give the reader some examples of data analysis of objectives where cluster analysis would be at an advantage or disadvantage.

**p.10.**, **Section 3.3**.: As you state, interpretability is key. Chemical interpretability is discussed in a concise way. However, in addition to chemical interpretation, looking at loading time series is an important indicator. Do the time series reflect the kind reaction kinetics (in experiment chamber) and take place in reasonable timescales?  This relates to e.g. Fig.13.

Comparing e.g. the PMF results in Figs. 13, S26, S27 – only the rank 3 Poisson-model (Fig S26-c) solutions' loadings (and factor 3 in Fig 13d) behave in a realistic way. The others anti-correlate highly, usually signaling they are over-resolved (unrealistically split) and usually then less components should be used. See for example f1 andf2 in Fig 13-a: correlation is undoubtedly close to -1 and the dynamics do not make sense – this simply seems like a bad factorization solution.

**p.13, Section 4.**: the factor profile figures (Figs 3 through 7) are really difficult to read this way! Please separate each factor to its own sub-plot, similar to Figure 12-b. Maybe put the fractional plots to supplementary material?

Figure 1, Figure 2. Please highlight in these figure factor numbers (e.g. a larger dot?) that were selected according to the evaluation metrics.

In all figures, for quick reference, please state if it is gas or particle (or AMS vs PTR-MS) data.

**p.14. Section 4.1.2**.: The difficulty to interpret PCA (negative) highlights that PCA separates principal components of variability, whether positive or negative, which can not [necessarily] directly be interpreted as physical [concentration] components of the system – the variability could be equally connected to losses of signal due to it chemically reacting away. Again, this ties to the objective of the analysis and should be discussed more clearly.

**p.17.l.25.** *"This is caused by the used error scheme, where errors are larger for the fast changes in the data (Fig. S4b)."* I am very confused. Is this "feature" intentional? This also relates to the question of p.7.l.15. Is the variability of ion concentration in chamber is indeed used as a metric for measurement uncertainty (even when scaled by sqrt(n))?. This seems a peculiar, to say the least, a very un-orthodox way to do error modeling in PMF. It could explain why PMF does surprisingly poorly compared to the other models for AMS data.

Importantly, please include a comparison of your error models versus "the standard" error model in-build in AMS data analysis software(s) (Squirrel / Pika / PET) and list the differences between the standard practice (see e.g. Allan et al., 2003; Ulbrich et al., 2009; Zhang et al., 2011) and your data pre-processing procedure.

**p.21., Section 4.2.4**.: I would have inspected the 2 factor solution, is it had percentage-wise the largest decrease in Q/Qexp. Usually it is also good practice to start from lower number that you can certainly interpret and continue to higher numbers.  This could explain also why EFA and PAM suggest 2 factors (clusters). Please show these 2 factor (cluster) solutions (in the supplementary material).

The analysis on the error scheme issues seems correct.

I have to disagree on the interpretability of the PMF solutions presented here. This solution seems a mathematical one rather than physically meaningful. See also comment to **p.10.**, **Section 3.3.**, Figure 13 etc. The time series indicate over-splitting of factors (extreme anti-correlation of f1,f2,f3) and the most of spectral profiles have extreme positive correlation (extreme high similarity by visual inspection). I don't really see many interpretable features in Figure 13. Mainly the HOA spectrum in Fig 13b, LV(f1,f2,f3)/SV(f4) split in 13 c&d.

**Technical corrections:**

**p.3.l.4.:** PTR generally refers to the ionization reaction and not the instrument, Please use PTR-MS or similar, common acronym.

**p.7.l.30**. $t_s$ here is sampling time per m/z channel, when scanning (Allan et al., 2003), not to be confused with the total sampling time of the instrument, so please add this clarification.

**p.21.l.13.:** Please add a reference to m/z 57 and 59 link to HOA.

**Supplementary material:** please check figure numbering is in numerical order. Similarly to main part figures, please add if data is PTR—MS or AMS.

*references*

Allan, J.D., J.L. Jimenez, H. Coe, K.N. Bower, P.I. Williams, and D.R. Worsnop, Quantitative Sampling Using an Aerodyne Aerosol Mass Spectrometer. Part 1: Techniques of Data Interpretation and Error Analysis, *Journal of Geophysical Research- Atmospheres*, Vol. 108, No. D3, 4090, doi:10.1029/2002JD002358, 2003. (NB! Includes corrigendum)

I.M. Ulbrich, M.R. Canagaratna, Q. Zhang, D.R. Worsnop, and J.L. Jimenez. Interpretation of Organic Components from Positive Matrix Factorization of Aerosol Mass Spectrometric Data. *Atmospheric Chemistry and Physics* , 9(9), 2891-2918, 2009.

P. Paatero and P. K. Hopke, "Discarding or Downweighting High-Noise Variables in Factor Analytic Models," Analytical Chimica Acta. Vol. 490, No. 1-2, 2003, pp. 277-289. http://dx.doi.org/10.1016/S0003-2670(02)01643-4

Q. Zhang, J.L. Jimenez, M.R. Canagaratna, I.M. Ulbrich, S.N. Ng, D.R. Worsnop, and Y. Sun. Understanding Atmospheric Organic Aerosols via Factor Analysis of Aerosol Mass Spectrometry: a Review. Analytical and Bioanalytical Chemistry, 401, 3045-3067, DOI:10.1007/s00216-011-5355-y, 2011.

---

## Author Comment (AC2) · 28 Mar 2020

**Response to reviewer 2, Mikko Äijälä**

We thank Dr. Äijälä for carefully reviewing our manuscript and providing very useful comments and suggestions, which improved our manuscript. Below we address each comment point by point.

5 For clarity we have marked the reviewer comment in **blue**, our answers in **black**, and changes to the manuscript in **red**. Page and line numbers (**in black**) in our replies refer to the clean revised manuscript (without tracked changes), and **green** line/page numbers refer to this response.

**Summary**

- 10 Sini Isokääntä and co-authors present a comparison of dimensionality-reductive techniques for mass spectral data analysis, applied to a data set of car exhaust + a-pinene aging experiment in a reaction chamber. The data, collected by a PTR-MS and an AMS, was factorized and clustered using 5 different techniques. The authors present and discuss their analysis procedures and interpretation of results from mathematical and physicochemical viewpoints. For the PTR-MS data, all five
- techniques produce comparable results, yielding 4 to 5 interpretable factors (or clusters). This is a very important result considering the novelty of applying these methods to PTR-MS and similar mass spectral data of gas phase. For AMS data, only PMF and NMF yielded results that could be compared PCA, EFA and PAM struggled with the particle data set. Although applied to data from a single experiment, the comparisons presented and the discussion on analysis techniques convey
   general messages for many similar analyses that abound in atmospheric science.

The manuscript is clearly of interest to AMT readers, and touches the important topic of computational analysis of complex, large mass spectrometric data sets and advanced statistical methods of data analysis. Despite the evident need, a very limited amount of reviews of advanced statistical methods for this type of complex, physicochemical analyses is available, so I see the authors' contribution as extremely welcome to the field. The manuscript is well and clearly written and structured. With some exceptions, detailed in specific comments, the data, analysis techniques and experimental setup is adequately described. I recommend the paper for publication in AMT after addressing the following comments.

**30**

**Major comments**

The authors generally do a good job of introducing and describing their methods, but I would call for discussion on the physicochemical objectives or purpose of the statistical techniques used in this study. Obviously it is about more than reducing the size or dimensions of the data, which is the

- 35 single common feature of all these SDRTs. However, some of these methods are rather similar (e.g. PMF vs NNMF) while some aren't (e.g. PAM vs factorization methods). Generally, a data analyst should choose a proper tool for the job, depending on the goal. Here it seems the methods are just applied on the data, seemingly without stating beforehand e.g. the difference in categorizing or classifying variables using PAM, studying the main explanative components of variability or doing
- 40 latent feature extraction by exploratory factorization. Many of these differences are later casually mentioned, but I suggest an effort to further summarize these fundamental differences, and conversely, some of the close similarities could be made in the introduction and conclusions of the paper.
- 45 We have now summarized and added discussion of the used SDRTs to a new section in the manuscript (section 4.4.), where we aimed to discuss the objectives and purposes of these techniques in more concise manner than mentioning these properties throughout the manuscript.

**4.4 Summary of the SDRTs used in this study**

The methods tested in this study have many similarities and many, fundamental or computational, differences between them. Though in literature, they are many times applied to similar problems. In this section we will summarize some of these properties.

[revised manuscript text omitted]

Taking account of everything above, the most important thing to consider when selecting the SDRT is the interpretation. What are the features the researcher wants to deduct from the data, what are the properties of the data and how the data can answer the research questions. As we have shown in the results section, the methods provide quite similar reconstructions of the time series, but the

- 15 interpretation of the steps leading to these are quite different. For example, comparing EFA factor FE4 in Fig. 3a and PMF factor FP4 in Fig. 7a reconstructions, they seem to show the same procedure but the first one includes both positive and negative effects and the second one consists only of positive effects.
- 20 Secondly: algorithmic methods are often discussed plenty in these types of reviews, but some of the equally significant, but less obvious issues include, among others: data pre-processing (e.g. scaling and weighting), metrics (e.g. for quality evaluation and similarity), and error models. Mostly, the authors cover these topics well enough, but some major questions surrounding e.g. the non-standard PMF error model the authors used and preprocessing of data for PCA/EFA are raised. See specific comments on the details.

In our manuscript we did use the standard AMS error model for PMF, and we understand that this was clearly not emphasized enough in the manuscript as our error description was misunderstood. The main point using the standard AMS error was indeed to show how "reference" error works for PME when compared to others. We have addressed this issue in detail in the specific comments (see

30 PMF when compared to others. We have addressed this issue in detail in the specific comments (see our reply to comment "p.7.1.26" in page 10) and modified the manuscript accordingly. We have also now named the "Poisson style error" to "standard AMS error" to emphasize this.

We have also clarified the data preprocessing steps for EFA and PCA as suggested, see our replies to specific comments "p.5.1.30" and "p.6.1.14" in pages 6 and 8, respectively.

Finally, most factorization methods, including those used in this paper, assume constant factor profiles, i.e. that chemical reactions do not take place. This is fundamentally at odds with the approach of using them to model data of reaction chamber chemistry. In practice, the methods can work despite this (as is shown in this work), but this fundamental violation of basic assumptions should be explicitly stated and discussed.

The assumption of constant factor profiles is indeed a very important one for PMF (and NMF). The idea that chemistry in the atmosphere/chamber violates this assumption is however a misconception stemming from a too narrow definition of what a factor represents. A factor can be seen as a direct (emission) source of compounds which changes its contribution to the whole signal (e.g. primary emissions from biomass burning as a fire develops and then dies). But a factor can also be interpreted as a group of compounds showing the same temporal behavior. If this group is released

together as an emission or if the compounds are formed in the same ratio by some chemical process should not matter. In the latter case, it is important how wide the group is selected, i.e. if we group products of processes together for which the contribution changes with time.

- As an example, consider a data set of gas phase measurements in the atmosphere. There is a mix of precursors and let there be OH and O3 chemistry during the day and NO3 and O3 chemistry at night. All of these reactions can form Highly Oxygenated Material (HOM) among other compounds. The ratio of the reaction pathways (and thus the observed HOM composition) will depend on the ambient conditions (e.g. O3 and NOx concentration, irradiation). If we do not constrain the PMF analysis, we will find more than one HOM-type factor (i.e. a daytime and a
- 10 night-time factor). Each of these factors then has a constant profile, only the contribution to the data set changes with time, i.e., the time series of the night-time factor will be close to 0 during the day and vice versa for the day-time factor. As there is no strict criterion for the correct number of factors, it is important that the scientist analyzing such a data set compares solutions with different number of factors and uses additional information (such as other measurements).
- 15 We also could perform a "rolling window" style PMF (looking at a short period of the data set at a time (e.g an hour) and then shifting this window by e.g. 10 min). If we set the number of factors to only find 1 HOM type factor in the whole data set, this factor would have a different factor profile during the day and at night. But this behavior was created by the choice of the scientist to have a certain number of factors.
- 20 In the analysis presented in our manuscript, we did not use any such assumption about the type of factors we were expecting or even their number. Rather we inspected the solutions for multiple factor numbers and chose the number of factors based on the presented criteria and the interpretability, i.e., we first ran the analysis and then interpreted the results. We have now stated the assumptions for constant factor profiles in the new section 4.4 with further discussion. See reply
- to the first major comment in pages 1 and 2.

**Specific comments:**

p.2.1.2.: "...sharp change at the beginning of experiments, e.g., switching on UV-lights) may present additional challenges, as PMF was originally developed for field measurement data sets
where real changes in factors are expected to be much slower than e.g. the noise in the data.". In my understanding PMF is not assuming anything about order of measurement points or rates of change in loadings – please provide a reference or expanded rationale for this line of thinking!

Indeed, PMF does not make any (mathematical) assumptions of the order of measurement points or rates of change. However, most of the PMF applications, so far, have been with ambient measurement data where, for most part of the data, the instrument noise is indeed more rapidly changing than the actual observed concentrations, which must be also assumed e.g. in the study by Ulbrich et al. (2009), where the AMS error scheme was introduced. We have now reworded the referred sentence to clarify that:

40

45

page 3, lines 5-8

[...] The special conditions in lab experiments (sharp change at the beginning of experiments, e.g., switching on UV-lights) present an additional test scenario, as PMF has been mostly used for field measurement data sets where the main focus is often in the long-term trends and real changes in factors are often expected to be more subtle, than e.g. the variations in the noise in the data. [...]

We also adjusted section 4.1.5 accordingly, please see reply to comment ("p.17.1.25") in page 15.

p.2.1.14. "In our study we chose a set of SDRTs having fundamental differences." Aside from data
reduction or simplification, the objective of SDRTs differ. Often, clustering is applied to classify or

categorize non-correlated variables, whereas factor analysis aims to uncover latent features the combination of which would best explain the variation in data. Can the authors please comment on their objective of the analysis outside of reduction of data size) of the use of SDRTs for this type of analysis? Especially regarding clustering.

5

We have added a new section 4.4 to the manuscript to include more discussion of the fundamental differences and objectives the various SDRTs might have. Please see our answer to the major comment in the beginning of this reply.

- 10 **p.2.1.16:** "On the other hand, clustering might be more suitable for a more simplified or preliminary approach, or when the chemical compounds in the data are already known." It is not trivial to envision a case where clustering would be preferable, even as a preliminary approach please expand on this reasoning or provide an example.
- 15 Please see our reply to specific comment "p.10.1.5." in page 13 for added examples. We modified the referred sentence to

section 1, page 3, lines 19-21

[...] On the other hand, clustering might be more suitable for a more simplified or preliminary
 approach (as it is computationally less demanding), or when the chemical compounds in the data are already known or if strict division between variables is preferred. [...]

**p.3.** Please describe the PTR-MS and the SP-AMS in some more detail here. Specifically their mass analyzer resolution (C/HR/L –ToF), as this strongly affects mass analysis and type of data.

25

Both of these instruments are extensively described in Kari et al. (2019) and in the references given there. However, we added the requested details of the mass analyzer resolutions shortly to the manuscript along with the other details requested by reviewer#1;

30 section 2, page 4 lines 13-22

[...] Detailed setup, calibration procedure and data analysis of the used high resolution PTR-MS have been explicitly presented in Kari et al., (2019b). In the campaign, the high mass resolution of the instrument (>5000) enabled the determination of the elemental compositions of measured VOCs. The instrumental setting was intended to minimize the fragmentation of most compounds, so

- 35 the quantitation of the VOCs was possible. The chemical composition of the particle phase of the formed SOA was monitored with a soot particle high resolution aerosol mass spectrometer (SP-AMS, Aerodyne Research Inc., USA, hereafter referred to only as AMS, Onasch et al., 2012). In brief, the SP-AMS was operated at 5 min saving cycles, switching between the electron ionization (EI) mode and SP mode. In EI mode, the V-mode mass spectra were processed to determine the
- 40 aerosol mass concentration and size distribution. The mass resolution in this mode reaches ~2000. The SP-mode mass spectra were used to obtain black carbon concentration. As the used chamber was a collapsible bag, the volume of the chamber decreased over time due to the air taken by the instruments. [...]
- **45 p.3.1.9.:** A note. As the authors state, if one does not account for mass spectra baseline, this may give rise to "background factors" or cause variables to get polluted or mixed (depending on instrument resolution). While theoretically these could be separated by factor analysis, in practice their presence in data hinders the analysis, often notably. For hard clustering, this effect of may even more problematic. Can you give an order of magnitude estimate of this effect? Can you

separate baseline factors / clusters in some cases? What is their fraction of total explained variability?

We were not able to separate, or at least qualitatively justify that the background factor, e.g. FE4

- 5 from EFA applied to the gas-phase data, would indeed be pure instrument and/or chamber background, as this factor also included ions related to the car exhaust, thus being slightly mixed. The lack of identified compounds for our gas phase measurements hinders the detailed analysis and identification of possible background factors. In principle, by adding more factors, we might be able to find one or more background factors, but as we would not have been able to identify those explicitly, we did not apply this approach here but rather acknowledged the possible uncertainty in
- 10 explicitly, we did not apply this approach here but rather acknowledged the possible uncertaint our factors. Therefore, also quantifying this effect here is impossible.

**p.3.1.18.:** This type of truncation correction is likely detrimental to analysis and should be avoided in statistical analysis (especially factor analysis, as it creates an artificial, non-random, variability component). If only few points had this problem, would it not be preferable to omit them?

The number of observations which needed to be elevated to small positive values was not large, and so was the number of time points Removal of one negative value from one variable would require the removal of the complete row, and as the negative values occur in a few variables, but not at the same time point, this would remove a significant amount of the data. The small negative values in the data are in cases when the concentration of the compound was in practice zero, but the application of the baseline correction caused them to be very small negative numbers instead. We acknowledge the possibility of increased uncertainty due to this kind of procedure, but due to the very small size of the AMS data set, omitting the negative data points was not possible. PMF can be

- 25 actually applied to some extent even with the existence of some negative values, but the NMF algorithm we used, does not allow the input of any negative values. In addition, also for PMF the data should be positive by definition and without extensive inspection of the source code of the PMF Evaluation tool we do not know how the software handles the negative numbers. Thus, to give all SDRTs the same data as an input, we replaced the negative values. To acknowledge this issue
- 30 for the AMS data more clearly, we modified the end of the section 4.2.5 (please also note that this section has now been modified to include the 2-factor solution from PMF instead of the 4-factor one)

page 24 lines 17-23

15

20

50

- 35 [...] This, on the other hand, assigns different emphasis between the matrices, possibly causing NMF to use more effort to reconstruct the data matrix with factor time series. However, the reader should keep in mind that for detailed chemical analysis of such a small data set, especially with PMF, the downweighing is advisable. In addition, the replacement of very small negative numbers with very small positive numbers is not mandatory for PMF, as it can run containing a few negative
- 40 values. However, we did the replacement here, as the NMF algorithm used here, does require strictly positive input data. Acquiring a balance between statistically good results and realistic factors might be challenging, and to achieve more robust results, testing different error schemes may be beneficial, especially for a data set with such a small size. [...]
- 45 **p.5.1.30:** "[...]where X includes the analysed variables (centred and scaled by their standard deviations)" In their examination of errors in PCA (and PMF) in environmental applications Paatero and Hopke (2003) strongly advice against "autoscaling":

*"5.1. Do not autoscale noisy variables in PCA* In PCA, it is customary to scale columns of X so that in the scaled matrix, all of the columns have the same variance. This procedure means that the sum of the two components of the variance (signal and noise) is constant over all variables. It follows that for the weakest variables, having the smallest amount of signal, the noise variance is much larger than for the strong variables. This behavior is in severe conflict with the recommendations found in this work: the exaggerated noise in a few noisy variables will cause the small principal components to be undetectable in the analysis and will increase the noise in other principal components. The recommendation is clear: "do not auto-scale noisy variables in PCA modeling" [...]

Is this type of scaling also done in here? Please reflect and add some discussion on the issue of error model in PCA. Perhaps recommend the readers to note this for future analyses?

The sentence to which the reviewer is referring, explains how the scores (factor time series) are usually calculated. However, in the end of the paragraph, we say "the scores are calculated without standardizing the data matrix to achieve more interpretable component time series".

15

35

40

45

5

However, we assume the paper from Paatero and Hopke means if the data matrix X is scaled before applying PCA (or inside the algorithm). We used 2 variations of PCA, EVD-PCA, which uses eigenvalue decomposition into the correlation matrix (calculated from unscaled X, though centering for example, would not have an effect on the correlations), and SVD-PCA, which uses singular

- 20 value decomposition into the data matrix X or the correlation matrix. Thus, no scaling, whatsoever, was applied when using EVD-PCA (which produced interpretable results shown in the manuscript). We also tested SVD-PCA, and applied it to the scaled data matrix, as this was the recommendation given for the algorithm we used. The SVD-PCA results were only shown in the supplement as they were not interpretable. However, as the reviewer explains, this might affect negatively into the
- 25 results and is contradicting to the recommendations given by Paatero and Hopke (2003). Therefore, we now also looked into the SVD-PCA applied to the unscaled data matrix X, but those results did not show any improvement. We added one example of this to the supplementary material (see section S4.1 Figure S13) and modified sections 4.1.2 and S4.1 accordingly. We modified sections 3.1.1 and 3.1.2 to emphasize that no scaling was applied to the data for EVD-PCA at any point and explain scaling/not scaling situation for SVD-PCA.

section 3.1.1

page 5 lines 23-25

[...] It should be noted, however, that Eq. (1) describes the theory behind the PCA model, not the actual calculation process, which is described below. Thus, for example, centering of variables is

not required. [...] page 5 lines 29-31

[...] In our study we applied EVD-PCA to the correlation matrix (calculated from the unscaled data matrix) and SVD-PCA was applied to the data matrix without and with the scaling (centered and scaled by their standard deviations). In addition, the acquired [...]

section 3.1.2, page 7 lines 6-8

[...] As for the PCA explained in the previous section, Eq. (4) here describes the form of an EFA model based on literature, not the direct calculation in the algorithm. Thus, no scaling (what e.g. the subtraction of the mean  $\bar{y}_1$  from  $y_1$  in Eq. (4) essentially is) is applied here. [...]

**section 4.1.2, page 16 lines 10-14**

[...] SVD-PCA (when applied to the scaled data matrix) was not able to separate  $\alpha$ -pinene as its own component, but instead created two factors which were dominated by the unreacted  $\alpha$ -pinene and its fragments (see Fig. S12 in SI section S4.1). In addition, the unrotated solution included a

7

large number of negative loadings, which complicated the interpretation of the components. No improvement was achieved when SVD-PCA was applied to the data matrix without any scaling (see. Fig. S13). [...]

**5 section S4.1, page 6 lines 14-17**

[...] components from SVD-PCA (applied to the scaled data matrix) and the unrotated component time series and original loadings (scaled eigenvalues) are shown in Fig. S12 with 4 components. The unrotated results from SVD-PCA when applied to the unscaled data matrix are shown in Fig. S13, and Fig. S14 shows the original loading values for the Oblimin rotated EVD-PCA with 4-components [1]

10 components. [...]

**p.5.1.24.** : While potentially simplifying the chemical interpretation of factors, rotating the variables in a direction where ions are more separated between the factors, does this not equally degrade the time series interpretability (loadings' time series get more similar)?

15

In PCA and EFA the rotation is applied to the loading matrix only, as the loading's time series are calculated after adjusting the actual loading values. This means in principle, that also the time series get more "different/separated" meaning rotation does not degrade, but in contrast even increase, the time series interpretability. In general, as pointed out, rotations potentially simplify the chemical interpretation of the factors, but they might decrease the overall fit, as the rotation is applied after

the solution has reached the optimal value for the condition to be minimized.

**p.6.l.14.:** Same question on EFA error model as for PCA above, as PCA and EFA are very similar methods. Please discuss error weighting and accounting signal-to-noise (Paatero and Hopke, 2003).

25

30

20

Please see also our replies above ("p.5.1.30" and "p.5.1.24"). Regarding the error weighing. In EFA (or in PCA) no kind of weighing is applied based on measurement error as it is done in PMF. We now noticed from our formulation for EFA, that we use the term "the error term  $\varepsilon_1$ ", which might cause misunderstanding. The term in question is not the error in the same sense as for PMF (which needs the error matrix), but a residual error arising from the model fit. We renamed it to "the residual term  $\varepsilon_1$ " to avoid further confusion.

Regarding the signal-to-noise, please see our reply to comment "p.7.1.21" in page 9.

**35 p.6.1.6.:** Does the limit of 0.3 apply to AMS data as well? This means any m/z signal explaining under 0.3 ug/m3 was set to zero? How many variables does this affect – I would imagine it is a very large fraction? How does this truncation affect mass (signal) conservation in data?

After applying PCA to the correlation matrix, the acquired eigenvectors were scaled by the square root of the eigenvalues (see page 5, lines xx-xx) to acquire similar loading values compared to EFA. The limit of 0.3 given here, was applied to those loading values, which are (and should be in most cases) between -1 and 1. It does not mean that signals explaining less than 0.3 µm/m3 are set to zero, but that the contribution of a variable into that specific factor is very small, and when setting the loading value to 0 for that factor, we enhance the separation between the factors. The limit of

0.3 for the absolute value of the loadings is widely applied in literature, and was thus selected (e.g. Field, 2013, Izquierdo et al., 2014). We adjusted the paragraph and references to justify our selection of the limit 0.3:

section 3.1.1, page 6 lines 26-29

[...] Very small loading values (absolute value less than 0.3) are suppressed to zero to enhance the separation of ions between the components. The limit of 0.3 was selected, as this is often given as a reference value for insignificant loadings (see. e.g. Field, 2013; Izquierdo et al., 2014). [...]

- 5 Regarding the mass signal conservation. As for EFA and PCA the factor scores can be calculated in various ways, and as the loading values can be negative, EFA and PCA do not conserve the original mass signal in that sense, but instead aim to conserve as much information (i.e. we could calculate how much of the correlations are preserved with different number of factors) from the original data as possible. Even though there are no practical obstacles to back-calculate the original signal from
- 10 the obtained loadings and scores (which are calculated with the loadings), it does not make sense in a similar manner as for NMF and PMF, where the "scores" and "loadings" are optimized at the same time inside the algorithm. Due to this, the concentration of the factors between these methods is not fully comparable, but this is not an issue in this type of analysis.
- **p.7.1.15** On the error model of st.dev / sqrt(n): as written in the text, st.dev reflects the changes in the concentrations of compounds in the chamber, during the experiment, and not measurement uncertainty (or counting statistics error of the instrument) in a repetition measurement? How does dividing it by (square root of) data series length make it a more relevant error metric? Please explain. Also: Why not use the standard error model computed by the AMS analysis software
   (Squirrel/Pika; e.g. Ulbrich et al., 2009)?

The PMF model was first introduced by using the standard deviations as an error metric, see Paatero and Tapper, (1994). The error metric applied here (st.dev / sqrt(n)) is commonly called "standard error". It takes into account that deductions from measurement with less observation

- 25 contain more uncertainty. As the standard deviation may be high in this type of data (containing the step change at the onset of photochemistry), it may be a too conservative estimate for uncertainty in PMF analysis. Accounting for the number of observations gives a more reasonable estimate for total uncertainty. We clarified this in section 3.1.3:
- 30 page 8 lines 8-11

35

[...] But as a reference, the standard error of the mean (the standard deviations of the ion traces divided by the square root of the number of observations, i.e., length of the ion time series) was used as an error for both PTR-MS and AMS data. It considers that measurements with less observations contain more uncertainty. These error values are constant for each ion throughout the time series [...]

We did use the "standard" AMS error in our study (named Poisson-like error). For more details see our reply to comment ("p.7.1.26") in page 10.

- 40 **p.7.l.21.** (and in the supplementary referred): In calculation of the signal-to-noise-ratio (SNR), you identified weak and bad variables (defined by their low SNR). However in contrast to usual practice (in PMF), you do not down-weight these signals. Especially since you are using a custom (not thoroughly validated) error model, I would listen by the advice, Paatero and Hopke (2003):
- "Regarding weak/bad variables, the main result of this work is that even a small
  amount of overweighting is quite harmful and should be avoided. In contrast,
  moderate downweighting, by a factor of 2 or 3, never hurts much and sometimes is
  useful. Thus, it is recommended to routinely downweight all weak variables by a
  factor of 2 or 3. This practice will act as insurance and protect against occasions
  when the error level of some variables has been underestimated resulting in a risk of
  overweighting such variables. Regarding bad variables (where hardly any signal is

**visible from the noise), the recommendation is that such variables be entirely omitted from the model."**

The reasoning for not down-weighting only states that low SNR data has high noise-to-signal rate, which is only stating the evident, and not very useful. Also, the small number of variables is not

- 5 really a good excuse to deviate from the practice (without at least some sensitivity analysis). While I agree the overall error from not doing this properly is likely smallish (for PMF), I recommend you acknowledge the issue and cite this "best practice", the Paatero and Hopke (2003) recommendation, for the readers – not to proliferate a deficient data pre-processing practice.
- 10 We have added more justification for omitting the downweighing. Please also see our reply to Reviewer#1 comment ("page22 line 20")

**section 3.1.3, page 8 line 27 – page 9 line 2**

[...]. In contrast to the suggested best practice (Paatero and Hopke, 2003), we did not

- downweighed any ions in our data sets. This approach was used in order to give each SDRT an equal starting point for the analysis, as e.g. for NMF or PCA similar downweighing is not possible, as we do not have any error estimates to calculate the signal-to-noise ratios in similar manner. However, to avoid misguiding the reader to omit recommended data pre-processing practice for PMF, we also tested PMF with downweighing. This, as expected, did not change our results
  significantly, but we acknowledge it should be indeed applied if aiming for a more detailed
- chemical interpretation of the PMF factors. [...]

**section S2.3, page 5, lines 1-5**

[...] However, no removal or downweighing of the variables was applied in the results presented in the main manuscript, as for the weak ions the error values were already in the range of the ion signal itself, and for the bad variables the error was usually way above the signal throughout the ion time series. In addition, the number of weak or bad ions for both AMS and PTR-MS data were rather small. To justify our decision, we did run the PMF analysis with the downweighing. This, as expected, did not change our results or interpretation, and thus those results are omitted. [...]

30

**Regarding the AMS error, see our reply to the next comment (to comment "p.7.1.26").**

**p.7.1.26.,** also Figure S4 (should be: "S7"?). The standard AMS error model is composed of a minimum (Gaussian) error (Ulbrich et al., 2009), related to electrical background noise of the

- 35 instrument (background at zero signal) plus a counting statistics uncertainty that follows the Poisson distribution (error is proportional to the square root of signal intensity; Allan et al., 2003). This model seems different from the ones used in this paper. Please comment on this, and why you chose not to directly use the approach of Allan et al and Ulbrich et al. (2009), readily available from the AMS analysis software.
- 40 From the plot S4 (should be "S7"?) "PMF error schemes" it seems the constant error scheme ("Static error") overestimates error, especially for low concentrations, and the counting statistics (Poisson) error seems negligible (underestimated), even for the most abundant aerosol ion, at m/z 44, at highest concentrations. Please double check and report the error calculations here against what you get from the "Squirrel" AMS analysis software.
- 45 Why was the "signal following error" not used/tested for the AMS, since it seems to work for PTR-MS?

We did use the standard AMS error model, as mentioned in our reply to major comment in page 3 line 27. We have now modified the manuscript accordingly to avoid this misunderstanding. In addition, we renamed the "Poisson stule error" to "standard AMS error" to emphasize this

50 addition, we renamed the "Poisson style error" to "standard AMS error" to emphasize this.

Section 3.1.3, page 8 lines 17-25

[...] For AMS, we also applied a standard error that is frequently used by the AMS community. The standard AMS error consists of the minimum error related to the duty cycle of the instrument and the counting statistics following the Poisson distribution (Allan et al., 2003; Ulbrich et al., 2009). Shortly, the standard AMS error for signal *I* can be formulated as

$$I_{err} = \alpha \sqrt{\frac{I_0 + I_C}{t_s}},\tag{9}$$

where  $\alpha$  is an empirically determined constant (here  $\alpha = 1.2$ , generated by the AMS analysis software PIKA, http://cires1.colorado.edu/jimenez-

group/ToFAMSResources/ToFSoftware/index.html, last visit on 9.9.2019),  $I_0$  and  $I_c$  are the raw signal of the ion of particle beam (ions s-1) for the chopper at open and closed position, respectively,

and  $t_s$  is the sampling time at a particular m/z channel (s). [...]

The signal following error was indeed tested for AMS data, but the results were omitted as they did not show any clear improvement and were rather similar to those acquired with the standard AMS error and thus would have lengthened the manuscript unnecessarily. We now mention this in the manuscript section 4.2.4:

page 23 lines 29-31

5

10

15

20

[...] The signal following error, used for PTR-MS, was also tested for particle phase data. However, as this type of error showed very similar behaviour as a time series as the Standard AMS error, and produced very similar outcome, those results are omitted from this manuscript. [...]

**p.8. l.6.:** How much does the Q (or Q/Q exp ratio) increase with these fpeak values?

Figure 1 below show the Q/Qexp for PTR-MS data with the 5-factor solution with static error (left) and signal following error (right). The change with fpeak is rather small, as seen from the figures. The actual factorization also does not change (for PTR-MS) with different fpeak, indicating the 5factor solution is quite robust.

30

Figure 1 Q/Qexp for PTR-MS data with the 5-factor solution with static error (left) and signal following error (right).

**p.9.1.10.:** How do you interpret the negative loadings in EFA and PCA? Why were these solutions deemed physically sound, if they feature negative mass loadings?

These negative loadings refer to the feature of EFA and PCA that the factor loadings can be negative. Negative factor loading may have different interpretations. It may indicate that the compound has a decreasing effect on the factor, i.e. it acts as a sink for the compounds with positive

- 5 loading in the same factor. In chamber experiments, negative loading may also refer to decreasing concentration of the compound within chemical reactions. For example, a compound that can be rather easily oxidized, or forms in the later state of the experiment, might get negative loading in a factor, that mostly contains later generation products. An example of this is benzene (detected as C6H7+), assigned to FE1. When inspecting the original loading values, it has a negative loading in
- 10 FE1 (identified as later/slowly forming products), and a positive loading in FE4 (identified as precursors from car exhaust/background). As benzene originates from the car exhaust, it contributes positively to FE4. However, as it oxidizes over the course of the experiment (thus it has a decreasing concentration), it contributes negatively to FE1, which mostly includes those later generation products.

**15**

20

We understand our approach does indeed conceal some additional information EFA/PCA can provide, but as we did not have enough detailed information of the measured compounds (e.g. the sum formulas were not identified for all compounds), the interpretation of the negative loadings separately, for the most part, was not possible. We added more clarification about this in the end of section 3.1.5, and also discuss this property of EFA/PCA in the new section 4.4 (see pages 1-3 in this reply).

**section 3.1.5, page 10 lines 9-19**

[...] However, this type of approach conceals the information of the negative factor loadings in
EFA and PCA (which are included in the calculation of factor/component time series as weights), but instead visualizes the general contribution of an ion to a factor. Negative factor loadings may have different interpretations. It may indicate that the compound has decreasing effect on the factor, i.e. it acts as a sink for the compounds with positive loading in the same factor. In chamber experiments, negative loading may also refer to decreasing concentration of the compound

- 30 participating in chemical reactions if it acts as a precursor for other compounds in the same factor. One example of this is benzene (C6H6), observed with PTR-MS. When inspecting the original loading values from EFA, for example, it has negative loading in FE1 (identified as later/slowly forming products), and positive loading in FE4 (identified as precursors from car exhaust/background). As benzene originates from the car exhaust, it contributes positively to FE4.
- 35 However, as it oxidizes over the course of the experiment (thus has decreasing concentration), it has a strong correlation with oxidation products but appears negative in FE1, which mostly includes those later generation products. [...]

p.10.1.5.: "Depending on the aim of the study 5 and the type of the data, this property of cluster
analysis may be considered either as an advantage or disadvantage.". This is an important point in this paper overall, as is the difference between hard vs soft divisions of variables. Please elaborate, and e.g. give the reader some examples of data analysis of objectives where cluster analysis would be at an advantage or disadvantage.

45 Obvious disadvantage is in cases where one compound affects multiple factors by physical or chemical definition. Then clustering loses information by forcing the compounds on only one cluster. For example, when identifying pollution sources, black carbon can be emitted from both wood burning emissions and traffic. There, clustering might not be the best approach. On the other hand, computational time (as shown also in this study) is one clear advantage for PAM or hard

division techniques in general, especially when analyzing very long ambient data sets. Hard division techniques have also been shown to work efficiently when distinguishing between different coffee types (i.e. strict separation between clusters is needed), as shown by hierarchical clustering by Sánchez-López et al. (2014).

5

We added few examples to section 3.2.1, page 11 lines 15-21

[...] One obvious advantage of cluster analysis (or hard division techniques in general) is computational time, especially if analyzing long ambient data sets. For laboratory measurements, this most likely is not an issue. Hard division techniques have also been shown to work efficiently for VOC measurements when distinguishing between different coffee types (espresso capsules), where strict separation between clusters is needed, as shown in Sánchez-López et al., (2014). For source apportionments studies, where one variable might originate from multiple sources, cluster analysis using hard division technique is probably not as suitable as softer division techniques,

which can assign one variable to multiple sources/factors. [...]

15

10

**p.10., Section 3.3.:** As you state, interpretability is key. Chemical interpretability is discussed in a concise way. However, in addition to chemical interpretation, looking at loading time series is an important indicator. Do the time series reflect the kind reaction kinetics (in experiment chamber) and take place in reasonable timescales? This relates to e.g. Fig.13.

- 20 Comparing e.g. the PMF results in Figs. 13, S26, S27 only the rank 3 Poisson-model (Fig S26-c) solutions' loadings (and factor 3 in Fig 13d) behave in a realistic way. The others anti-correlate highly, usually signaling they are over-resolved (unrealistically split) and usually then less components should be used. See for example f1 andf2 in Fig 13-a: correlation is undoubtedly close to -1 and the dynamics do not make sense this simply seems like a bad factorization solution.
- 25

30

35

Following the suggestion regarding the PMF solution for AMS in comments below (see reply to comment "p.21., Section 4.2.4" in page 15) we now present a 2-factor solution in the PMF results. The figure below shows the evolution of the two PMF factors and  $\alpha$ -pinene. In the chamber, we assume a steady-state OH concentration of 107 molecules/cm3, and the reaction rate constant OH- $\alpha$ -pinene of 52.3×10-12 cm3 molecule-1 s-1, it is very likely to get  $\alpha$ -pinene consumed (25ppb) in a time scale of half an hour. If we assume a mass yield of a-pinene SOA of 30% in this study (Kari et al., 2019), the consumed  $\alpha$ -pinene (about 140 µg m-3) can be transformed to a SOA mass concentration of 52 µg m-3. This is consistent with what is observed in the figure. Thus, we can assume the time series from the SDRTs used in this study reflect the reaction kinetics in the chamber and the processes take place on reasonable timescales.

Figure 2 Two-factor solution for particle-phase data from PMF (fpeak = 0) with the injected  $\alpha$ -pinene.

**p.13, Section 4.:** the factor profile figures (Figs 3 through 7) are really difficult to read this way! Please separate each factor to its own sub-plot, similar to Figure 12-b. Maybe put the fractional plots to supplementary material?

Figure 1, Figure 2. Please highlight in these figure factor numbers (e.g. a larger dot?) that were selected according to the evaluation metrics.

In all figures, for quick reference, please state if it is gas or particle (or AMS vs PTR-MS) data.

Figures 1, 2 and 11 are now modified as suggested, RSS in Figure 2 and 11 is changed to more NMF related metric as suggested by Reviewer #1. Captions in each figure were adjusted to include the information of the data (AMS or PTR-MS), and we added factor description (those in table 1 & 2) to the figures in the manuscript (when applicable), as requested by the reviewer#1.

With the factor contribution figures,

---

## Author Comment (AC1)

**Response to reviewer 1**

We thank the Reviewer for carefully reviewing our manuscript and providing insightful comments. Below we address each comment point by point. For clarity we mark the reviewer comment in **blue**, our answers in **black**, and changes to the manuscript in **red**. Page and line numbers (in **black**) in our replies refer to the clean revised manuscript (without tracked changes), and **green** line/page numbers refer to this response.

**Summary:**

5

30

- 10 This work accomplishes a cross-comparison of several data-reduction analysis techniques. The analysis is performed on a chamber experiment. Car exhaust was directly sampled into an environmental chamber. A-pinene was also added to the chamber. The mixed car exhaust/pinene was then aged via OH-initiated photooxidation. Two instruments were used, a PTR-ToF MS and an AMS. Data were then analyzed using principal component analysis, positive matrix factorization,
- 15 exploratory factor analysis, clustering, and non-negative matrix factorization. The resulting simplifications are compared in terms of the ability to reconstruct the original data set (residual), number of factors/groups required to explain data variability, time-series behavior of the factors, and chemical composition of the factors. The authors find that the preferred number of factors is roughly similar regardless of technique. Some factors (or groupings) are generally consistent (in
- 20 terms of time-series behavior and composition) regardless of technique. Some techniques, particularly PCA and EFA were found to be difficult to interpret and not as useful. The clustering method was not useful for AMS data.

**Major comments:**

25 1) The pre-light mixing period is a substantial fraction of the whole experiment- is this interesting? The inclusion of this time period seems to have significant impact on the algorithm results.

One of our aims was to investigate the robustness of the different SDRTs with a data set having large changes in the concentration. The mixing period was important to include in the analysis as the step change when the photochemistry is started in the chamber is a good test scenario because we know the reason for the change and the exact time. Also, the direction of change is known. Please see our reply to specific comment ("page 7 line 16") in page 4.

- 2) Mostly this paper seems to show that various different algorithms distinguish similarly between
  primary and secondary VOCs. Is this a useful reduction of chamber data, to group all secondary VOCs into one or two blocs? From Table 1 it seems that the two matrix factorization techniques result in perhaps three oxidized factors, but this is hardly discussed in the text. It is not clear to me if the three oxidized factors are consistent between the two techniques. It would be helpful to have more interpretation of these factors, especially if it were supported by a more detailed
- 40 connection to the chemistry of the system.
   NMF also seems to result in some mixing of primary and secondary emissions (FN4) which is probably unphysical.
- We want to point out that the PTR-MS mainly provides information on the precursors and not the heavily oxygenated compounds created in their oxidation. Thus, the data set is far from giving a full picture of the chemistry in the chamber during photo oxidation (see also our reply to specific comment "Page 24 line 13-14" in page 10). Separating primary emissions and grouping the reaction products into three groups without adding any additional information about the ongoing reactions is the first step for understanding the chemical processes in these experiments. For a more detailed study, gas phase data from different mass spectra that cover a wider range of oxidation products

(e.g. PTR-MS and I- CIMS) need to be combined. But before this can be attempted, we wanted to investigate the performance of the available SDTRs which to our knowledge has not been performed in such detail yet. However, we added the factor contributions as separate spectra to the SI section S5 to clarify the similarity of the acquired results of the different SDRTs used (Figures

5 S21 and S22) and added a few more comments there in addition to the more "mathematical" comparison of the factors by contrast angle.

**S5, page 13, lines 16-20**

- [...] Figure S21 and S22 show the factor contributions in separate panels for each factor for the main results from gas phase data (PTR-MS). Despite the fundamental differences between these methods, the relative factor contributions clearly show similarity. Even PAM, that is relatively different when compared to other SDRTs as it only assigns one m/z into one factor, the same ions are enhanced when compared to other SDRTs. [...]
- 3) The way the mass spectra (chemical composition of groupings) are presented is very difficult to
  interpret and compare. From Table 1 the authors have assigned a consistent identity to factors resulting from each technique. Can you show a direct comparison of "Factor 2" for example, perhaps by plotting one mass spectra against the other so that it is easy to see which VOCs are similarly enhanced, and which may be different?
- 20 We now added figures with the factor mass spectra in separate panels in the SI material section S5 for easier comparison (Figures S21 and S22). In addition, the contrast angle -procedure introduced in SI section S5 shows a direct mathematical way to compare the factor spectra between the different SDRTs.

**25 Specific comments:**

Page 3, line 3-4: I disagree with this statement, "PMF was originally developed for field measurement datasets where real changes in factors are expected to be much slower than e.g. the noise in the data." In the ambient environment, VOC composition and concentrations can actually
change very quickly (on the order of seconds), especially when plumes of highly-concentrated primary emissions are intercepted. The abrupt changes in conditions during a chamber experiment therefore do not present a special challenge to PMF, compared to ambient measurements.

We agree with the reviewer that also in ambient conditions, the changes can be fast depending on the environment. Hence our statement was misleading. We wanted to highlight here with our statement, that most of the PMF applications, so far, have included ambient measurement data. We have now reworded our claim into:

page 3, lines 5-8

- 40 [...] The special conditions in lab experiments (sharp change at the beginning of experiments, e.g., switching on UV-lights) present an additional test scenario, as PMF has been mostly used for field measurement data sets where the main focus is often in the long-term trends and real changes in factors are often expected to be more subtle, than e.g. the variations in the noise in the data. [...]
- We also adjusted section 4.1.5 accordingly: page 19, lines 17-24
  [...] This is caused by the used error scheme, where errors are larger for the fast changes in the data (Fig. S4b). In ambient data not measured at instant proximity of strong emission sources, for which PMF is often used, this type of error is beneficial as there the fast changes are more likely to be
- 50 noise or instrument malfunctions (excluding, for example, sudden primary emission plumes), and

we are more interested of the long-term changes instead. For laboratory data, where large changes are often caused by rapid changes in actual experimental conditions, e.g. due to injecting  $\alpha$ -pinene or turning the UV lights on, the static type of error is most likely preferable. Usage of the static error scheme helps to avoid overcorrecting intentional (large) changes in experimental conditions

5 and confusing them with real variation taking place during the experiment and typically being much less pronounced. [...]

Page 3, lines 26-29: A few more details are needed here (instead of in the supplement): What was the typical concentration of total VOC (or of a few key VOC e.g. aromatics)? What concentration of  $\alpha$ -pinene was added?

What concentration of α-pinene was added?
 What was the VOC-NOx ratio, and how was it adjusted?
 Why were these specific concentrations of vehicle exhaust VOC and α-pinene chosen?
 Were vehicle exhaust and pinene added just at the beginning of the experiment, or were they continuously injected?

15 Was the chamber continuously refilled (and with clean air or with fresh emissions?) to replace air taken by the mass spectrometers, or did the volume of the chamber decrease over time?

We aimed to keep this description as brief as possible as the prime focus of this manuscript was the comparison of the SRDTs. Also, the experimental conditions for the whole measurement campaign are described and interpreted in detail in Kari et al. (2019). However, we adjusted section 2 to include the requested information, and also the mass resolution details requested by reviewer#2:

page 3 line 31 – page 4 line 4

20

[...] The exhaust was diluted using a two-stage dilution system and fed into the 29 m3 collapsible environmental PTFE chamber ILMARI (Leskinen et al., 2015). For the experiment investigated in

- 25 environmental PTFE chamber ILMARI (Leskinen et al., 2015). For the experiment investigated in this study,  $\alpha$ -pinene (~ 1  $\mu$ L, corresponding to 5 ppbV) was injected into the chamber to resemble biogenic VOCs in typical suburban areas in Finland. Atmospherically relevant conditions were simulated by adding O3 to convert extra NO from vehicle emissions to NO2 and adding more NO2 to the chamber if needed. With these additions, atmospherically relevant VOC-to-NOx (~ 7.4
- 30 ppbC/ppb) and NO2-to-NO ratios were achieved to resemble the typical observed level in suburban areas (National Research Council, 1991). [...]

page 4, lines 10-22

[...] Volatile organic compounds (VOCs) in the gas phase were monitored with a proton-transferreaction time-of-flight mass spectrometer (PTR-TOF MS 8000, Ionicon Analytik, Austria, hereafter

- 35 referred to as PTR-MS). Typical concentration for few example VOCs in the mid-way of the experiment were 2  $\mu$ m/m3 for toluene, 0.2  $\mu$ m/m3 for TMB (trimethylbenzene) and 1.7  $\mu$ m/m3 for C4H4O3. Detailed setup, calibration procedure and data analysis of the used high resolution PTR-MS have been explicitly presented in Kari et al., (2019b). In the campaign, the high mass resolution of the instrument (>5000) enabled the determination of the elemental compositions of measured
- 40 VOCs. The instrumental setting was intended to minimize the fragmentation of most compounds, so the quantitation of the VOCs was possible. The chemical composition of the particle phase of the formed SOA was monitored with a soot particle high resolution aerosol mass spectrometer (SP-AMS, Aerodyne Research Inc., USA, hereafter referred to only as AMS, Onasch et al., 2012). In brief, the SP-AMS was operated at 5 min saving cycles, switching between the electron ionization
- 45 (EI) mode and SP mode. In EI mode, the V-mode mass spectra were processed to determine the aerosol mass concentration and size distribution. The mass resolution in this mode reaches ~2000. The SP-mode mass spectra were used to obtain black carbon concentration. As the used chamber

was a collapsible bag, the volume of the chamber decreased over time due to the air taken by the instruments. [...]

Section 3.1.2: EFA seems very similar to PMF. Could you please explain the major relevant difference(s) between EFA and PMF, and how they would affect the resulting dimensionality reduction?

The main difference between EFA and PMF is that in EFA, the factorization algorithm is applied to the correlation matrix constructed from the data, whereas in PMF, the original data matrix is factorized. This means, that in EFA, the created factors include variables, that have stronger correlation compared to variables from other factors.

In PMF, the error matrix is used as a tool to downweigh noisy (or weak) signals. This can also be used to put emphases on certain parts of the data set (e.g. the part with most change in signal). EFA, however, has other options to discard noisy or insignificant signals (see last paragraph in section 3.1.1).

We have now added a summary section (new section 4.4.) to address and discuss the similarities and differences between the presented SDRTs as requested here and by reviewer#2.

**20**

5

10

15

Page 7 line 16 (and elsewhere): At multiple points in the manuscript it is mentioned that the rapid changes associated with lights-on cause problems when implementing the various dimensionality reduction techniques. Would it make more sense to exclude data prior to t=0? Was there a reason this was not done?

- 25 For a detailed analysis of such a chamber experiment, the data before the onset of photo chemistry would indeed be omitted. But the main focus of our study was to compare the performance of the different SDRTs with different types of mass spectra data. As the reviewer points out above, rapid changes in composition can also occur in the atmosphere. Thus, it is important to test the response of all SDTRs to such a change. The induced step change when photochemistry is started in the
- 30 chamber is a good test scenario as we know what caused the change and the exact time. Also, the direction of change is known (precursor getting consumed and products formed).

We added a few sentences to the manuscript to further justify our selection of the data:

35 section 1, page 3, lines 24-25

[...] Further, we examine the performance of the SDTRs when the data includes large and rapid changes in the composition. [...]

section 2, page 5, lines 2-4

- 40 [...] In addition, as the main focus of our study was to compare the performance of the different SDRTs with different types of mass spectra, instead of detailed analysis of the chamber experiment, we have also included the pre-mixing period during the  $\alpha$ -pinene injection (i.e., t < 0) into our analysis. [...]
- 45 Page 10 lines 28-31: Since Figure 1 relies on a comparison of BIC and SRMR, it would be helpful here to provide more detail on how these two metrics are calculated, what the relevant differences are, and why one may be preferred over the other. What was the purpose of calculating both metrics and why were these particular metrics chosen?

These particular metrics were selected, as they measure slightly different properties of the model. BIC is a comparative measure of fit balancing between increased likelihood of the model by adding parameters and a penalty term for number of parameters. The SRMR is an absolute measure of fit and is defined as the standardized difference between the observed correlation and the predicted correlation. It is a positively biased measure and that bias is greater for small N and for low df (degrees of freedom) studies.

The definitions and equations of these metrics can be now found from the SI material (new section S3.2), and we added a few sentences to manuscript section 3.3, as requested (page 12, lines 12-15):

10

5

[...] These metrics measure slightly different properties of the model. BIC is a comparative measure of the fit, balancing between increased likelihood of the model and a penalty term for number of parameters. The SRMR is an absolute measure of fit and is defined as the standardized difference between the observed correlation and the predicted correlation. See S3.2 for more details. [...]

15

Equations 11 and 12: There are several errors in these equations which are likely a copying error from Brunet et al. 2004. The authors should check that the actual implementation was done correctly. The equations should read:

$$H_{au} \leftarrow H_{au} \frac{\sum_{i} W_{ia} X_{iu} / (WH)_{iu}}{\sum_{k} W_{ka}}$$

20

45

$$W_{ia} \leftarrow W_{ia} \frac{\sum_{u} H_{au} X_{iu} / (WH)_{iu}}{\sum_{v} H_{av}}.$$

I also suggest here to use "k" as the row index for W (in the denominator term), to avoid confusion with "p" being the factor rank, and for consistency with Lee and Seung, 2001.

25 We thank the reviewer to pointing this out. But while the formula was incorrect in the manuscript text, all calculations we performed with the correct set of equations. Thus, none of the results for NMF need to be changed. The equation is now corrected, and the index "p" is changed to "k" as suggested. We added a sentence before equation (11) to clarify the connection between k and p:

**30 page 9, line 20**

Page 11 lines 29-33: Given that the update functions (11) and (12) are derived from the divergence cost function *D*(*X*||*WH*) (Lee and Seung, 2001, Eq. 3), I suggest that this cost function is monitored as a function of p, analogously to Q/Qexp(p) for PMF. The termination condition for NMF wasn't described in Section 3.1.4, but presumably it is not dependent on p; if this is the case then the divergence of the end solution can be compared for each value of p. The residual sum of squares is not an appropriate metric, as this was not the cost function used for the NMF implementation.

40 The update functions presented in the manuscript are indeed derived from the Eq. 3 presented in Lee and Seung (2001). We have now modified the manuscript and use the value of the cost function in the last iteration for each p instead of the RSS, as suggested.

We modified the last paragraph of section 3.3, corresponding figures (Figure 2 and Figure 11), and the results (sections 4.1.4 and 4.2.3) accordingly:

section 3.3 page xx, lines xx-xx

[...] The value of k is equivalent to the selected factorization rank p. [...]

[...] In addition, we investigated the cost function that approximates the quality of factorization as a function of the factorization rank p. For the brunet-algorithm that we applied in this study, this cost function is the divergence between data matrix X and the approximation WH (see Eg. (3) in Lee and Seung (2001)). [...]

5

10

section 4.1.4

page xx, lines xx-xx

[...] Figure 2a shows the divergence of the cost function D(X||WH) and CCC for factorization ranks from 2 to 10 for NMF. The CCC has a first decrease in the values at the rank 4 and the D(X||WH) shows an inflection point around the ranks 4-5. Figure 6 shows the factor time series and total contribution for the NMF with factorization rank 5. Five factors were selected, even though CCC suggest only 4 factors, as [...]

section 4.2.3

15 page xx, lines xx-xx

[...] The D(X||WH) has an inflection point at factorization rank 4 and CCC shows the first decrease in the values with 4 factors, as shown in Fig. 11a. [...]

Page 12 line 2: What is meant by "not achieved only by change?"

20

The purpose of the resampling here was to approximate and reduce the uncertainty in the factorization, based only on one dataset. As the figures show, the results vary significantly, especially when fast changes are taking place. This points out that all methods are rather sensitive for only small changes in the data and thus this kind of sensitivity test is necessary.

In order to avoid confusion by wording, the first sentences of section 3.4 (page 13, lines 20-23) were reformulated to:

[...] When analyzing the datasets, we realized that all of the factorization methods in this study are sensitive to even small changes in the data. In order to cross-validate the calculated factorization
 and approximate the uncertainty in the factors, 20 resamples of the measurement data were created with bootstrap-type sampling (Efron and Tisbshirani, 1986), i.e., sampling with replacement from the original data. [...]

Page 13 line 5: Why not compare the absolute value of the residual?

35

For PMF and NMF, which follow the matrix equation X = WH + E, where E is the residual matrix, this type of comparison is applied in by calculating the total residuals as a function of time (i.e. summing all variables for each time point in E), see e.g. page 24 lines 11-12 for AMS. However, as described in section 3.4 (see page 14, lines 16-25), this type of residual calculation is not possible for EFA and PCA, as they factorize the correlation matrix and do not create W and H in a similar

- 40 for EFA and PCA, as they factorize the correlation matrix and do not create W and H in a similar manner (i.e. simultaneously). Therefore, we can calculate the residuals for EFA/PCA only by reconstructing the correlation matrix (these values are shown in the manuscript, e.g. for EFA in sect. 4.1.1, page 15, line 24), but not the actual data. Thus, the actual residual values between PMF/NMF and PCA/EFA are not comparable. In addition, in PAM, which uses the distance matrix in the "decomposition" the situation is again very different.
- 45 in the "decomposition", the situation is again very different.

Pages 14 and 15: For other researchers which would like to use this paper as a guide, it would be helpful to indicate the range of values that are acceptable. For example Page 14 line 4-5, what value of residual would be considered not acceptable? Page 14 line 32, what is the Kaiser limit and what

range of values are considered "close"? Page 15 line 11 are these considered large or small residuals?

As desirable as such absolute values would be, unfortunately it is not possible to set them in a general manner. Especially for PMF and NMF, there is a strong subjective element in the selection 5 of p values and in the magnitude of acceptable residuals. It may depend on the type of data and the purpose of the analysis what is considered acceptable. The values may differ between analyzing a long ambient data set for the possible SOA sources or types (e.g. OOA vs HOA in an AMS data set) and studying a chamber data set trying to identify chemical processes. Please also note that most of these single value "goodness of fit" parameters are summarized over all observations and variables. 10 It is known that for PMF the overall Q values may be low while Q values for individual variables are not (interactive comment from Paatero to Yan et al., 2016).

The Kaiser limit/criterion refers to the ideal value of 1, and components having lower eigenvalues 15 should be discarded (explained in sect. 3.3, page 12, lines 6-7). What is meant by being "rather close" to this limit, is that the relative decrease of the eigenvalues is getting smaller and smaller with increasing number of components. So even though we reach the Kaiser limit (our eigenvalues get smaller than 1) having as many as 9 components, it is not realistic to have this type of system with that many components. In addition, as mentioned in sect. 3.3 page 12, lines 17-20, this type of tests for the number of components/factors are not meant to be taken as strict rules, but rather give 20 the first insights into the possible dimensions of the investigated system. We determined the most suitable number of dimensions for each data after investigating multiple solutions from various SDRTs.

- 25 As explained in sect. 3.4 (page 14, lines 27-29) the theoretical maximum value for the mean and IQR of the absolute residual correlations is 1 (indicating in principle no reconstruction at all) and the minimum 0 (perfect reconstruction). For example, if we would have a correlation coefficient of 0.6, and the absolute residual correlation is 0.0116 (as given the mean value in the line referred for PCA), we over- or underestimate the correlation coefficient by 1.93 % ((0.0116/0.6)\*100). Thus,
- the smaller values we have, the better the reconstruction. Unfortunately, there are no general 30 guidelines, and our main aim here was to compare if the reconstruction differs between EFA and PCA. To give the reader a more concrete way to understand the actual "size" of the errors for EFA/PCA in this study, we added the similar example to sect. 3.4:

35 page xx, lines xx-xx

[...] For example, for a variable-pair having a correlation coefficient of 0.7, a mean absolute correlation residual of 0.02, and an IQR of 0.04, this would mean the model over- or underestimates the correlation by 2.86% (= (0.02/0.7) \* 100). An IQR of 0.15 would mean that 50 % of all variable-pairs with correlation of 0.7 are within 5.7% (= (0.04/0.7) \* 100) of the original value of 0.7. [...]

40

Page 14 line 1 and page 15 line 2: Can you show please how it is determined that the additional component is not a new component with different properties but rather a mixture of previous components?

Figure 1 below shows the related factors (FE5 from the 5-factor case and FE2&FE4 from 4-factor 45 case) for EFA. From the time series we can see that the FE5 has a small peak at t = 0, suggesting it has picked up some properties from FE2. For example, m/z 89.06 in FE4 in the 4-factor case (Fig. 2b, contribution of 0.74 in FE4, others < 0.16), is divided more "equally" between the factors when using 5 factors (contribution of 0.17, 0.04, 0.11, 0.42, 0.26 for factors FE1-FE5), thus not being as

well separated. Similarly, some other m/z are more scattered between the factors in the 5-factor case, whereas with 4 factors they are more or less assigned to one, or possibly to 2, factor(s) only. This indicates the addition of the 5th factor does not introduce new properties but rather splits the existing ones into subsets. The reasoning for PCA is similar.

Figure 1 Factors 2 and 4 from the 4-factor case and factor 5 from the 5-factor case. (a) shows the time series and (b) the factor contribution.